# FAM76B regulates NF-κB-mediated inflammatory pathway by influencing the translocation of hnRNPA2B1

**Dongyang Wang[1,2†], Xiaojing Zheng[1†], Lihong Chai[1], Junli Zhao[1], Jiuling Zhu[1], Yanqing Li[1], Peiyan Yang[1], Qinwen Mao[3]\*, Haibin Xia[1]\***

[1]Laboratory of Gene Therapy, Department of Biochemistry, College of Life Sciences, Shaanxi Normal University, Xi'an, China; [2]Translational Medicine Center, Northwest Women's and Children's Hospital, Xi'an, China; [3]Department of Pathology, University of Utah, Salt Lake, United States

**Abstract** FAM76B has been reported to be a nuclear speckle-localized protein with unknown function. In this study, FAM76B was first demonstrated to inhibit the NF-κB-mediated inflammatory pathway by affecting the translocation of hnRNPA2B1 in vitro. We further showed that FAM76B suppressed inflammation in vivo using a traumatic brain injury (TBI) mouse model. Lastly, FAM76B was shown to interact with hnRNPA2B1 in human tissues taken from patients with acute, organizing, and chronic TBI, and with different neurodegenerative diseases. The results suggested that FAM76B mediated neuroinflammation via influencing the translocation of hnRNPA2B1 in vivo during TBI repair and neurodegenerative diseases. In summary, we for the first time demonstrated the role of FAM76B in regulating inflammation and further showed that FAM76B could regulate the NF-κB-mediated inflammatory pathway by affecting hnRNPA2B1 translocation, which provides new information for studying the mechanism of inflammation regulation.

**\*For correspondence:**
Qinwen.Mao@path.utah.edu (QM);
hbxia2001@163.com (HX)

†These authors contributed equally to this work

**Competing interest:** The authors declare that no competing interests exist.

## Editor's evaluation

This fundamental study identifies the protein FAM76B as a regulator of inflammation and provides evidence for its mechanism of action. FAM76B previously had no clear cellular function but the solid experimental methods described in this study established it as a negative regulator of inflammation. Additional investigation into the FAM76B protein is needed to better understand its role in physiological and pathological conditions associated with inflammation.

## Introduction

Peripheral immune cells mediating inflammation have been reported to be closely associated with the development of some diseases, such as cancer (*Moore et al., 2010*; *Kay et al., 2019*; *Suarez-Carmona et al., 2017*; *Khansari et al., 2009*; *Khandia and Munjal, 2020*; *Kolb et al., 2016*), obesity (*Kawai et al., 2021*; *Khodabandehloo et al., 2016*; *Seong et al., 2019*; *Saltiel and Olefsky, 2017*; *Curley et al., 2021*), and autoimmune diseases (*Kumar, 2019*; *Xie et al., 2019*; *Lochhead et al., 2021*; *Venkatesha et al., 2016*; *Abou-Raya and Abou-Raya, 2006*), among others. Microglia, as the resident macrophages of the central nervous system (CNS), act as the first line of defense in the brain (*Muzio et al., 2021*) and play a role in mediating neuroinflammation. It has been demonstrated that neuroinflammation is an important characteristic of almost all neurological disorders (*Gilhus and Deuschl, 2019*; *Brambilla, 2019*), which is the common thread that connects brain injuries to neurodegenerative diseases, and researchers have provided evidence of this commonality using traumatic

brain injury (TBI) as an example. Inflammation is regulated by different signaling pathways (*Kumar et al., 2003*; *Kim et al., 2004*; *Rawlings et al., 2004*; *Hawkins and Stephens, 2015*; *Lawrence, 2009*); however, the detailed mechanisms regulating inflammation are still poorly understood.

Human FAM76B is a 39 kDa nuclear speckle-localized protein that consists of 339 amino acids. It contains homopolymeric histidine tracts that are considered as a targeting signal for nuclear speckles (*Alvarez et al., 2003*; *Herrmann and Mancini, 2001*; *Salichs et al., 2009*). Although the function of FAM76B is still unknown, many poly(His)-containing proteins have been shown to be involved in DNA- and RNA-related functions and are overrepresented during the development of the nervous system (*Salichs et al., 2009*). In our previous study, by using immunohistochemical staining with custom-made anti-human FAM76B monoclonal antibodies (*Zheng et al., 2016*), we found strong immuno-labeling of FAM76B in the human brain, lymph nodes, and spleen. The results raised the question: does this protein play a role in regulating inflammation and neuroinflammation? In this study, we for the first time demonstrated that FAM76B could inhibit inflammation in vitro and in vivo by inactivating the NF-κB pathway. Furthermore, we showed that FAM76B could regulate the NF-κB pathway and mediate inflammation by affecting the translocation of hnRNPA2B1.

## Results

### FAM76B plays an anti-inflammatory role in macrophages in vitro

FAM76B is a nuclear speckle-localized protein with previously unknown function. We previously found that FAM76B was highly expressed in U937 cells using an in-house FAM76B antibody (*Zheng et al., 2016*). U937 is a human macrophage cell line that has been widely used to study inflammation in vitro. Therefore, we hypothesized that FAM76B might involve inflammation in U937 cells in this study. To test the hypothesis, we first produced a *FAM76B* knockdown U937 cell line (*Fam76b* KD) by lentivirus-mediated Cas9/sgRNA genome editing. The expression of FAM76B in U937 cells with *Fam76b* KD was confirmed by western blot (*Figure 1a*). Following treatment with PMA plus LPS/IFNγ, the cell line showed markedly increased expressions of *IL6*, *PTGS2*, *TNFα*, and *IL10* (*Figure 1b*). The increase in *IL6* expression in the *Fam76b* KD cell line was more prominent than the increases in *PTGS2* or in *TNFα* expression. The results indicated that FAM76B was involved in regulating inflammation in U937 cells. Furthermore, a *FAM76B* gene knockout (*Fam76b-/-*) U937 cell line was obtained using Cas9/sgRNA technology followed by drug screening and dilution cloning, then was confirmed by western blot and sequencing (*Figure 1c* and *Figure 1—figure supplement 1*). Similarly, significantly increased *IL6* expression was observed in the PMA+LPS/IFNγ-treated *Fam76b-/-* cell line (*Figure 1d*). Moreover, lentivirus-mediated overexpression of FAM76B rescued the function of FAM76B and reduced cytokine mRNA levels in *Fam76b-/-* cells (*Figure 1e and f*). These results demonstrated that FAM76B could inhibit inflammation in macrophages in vitro.

### FAM76B regulates the NF-κB pathway via influencing the translocation of hnRNPA2B1 protein

The data above showed that FAM76B had an anti-inflammatory effect by suppressing the expression of proinflammatory cytokines, especially IL-6. IL-6 is one of the important mediators of the inflammatory response, and it can be activated by NF-κB. NF-κB/IL-6 signaling has long been considered as a major proinflammatory signaling pathway in the peripheral tissues and brain. To explore the mechanisms of FAM76B regulating inflammation in U937 cells, the luciferase reporter vector of the *IL6* promoter was constructed. HEK293 cells transfected with this vector showed upregulated activity of the *IL6* promoter by NF-κB overexpression, confirming the activity of the reporter vector (*Figure 2a*). Interestingly, *FAM76B* knockout in HEK293 cells further increased the promoter activity of *IL6*, compared to wild-type (WT) cells (*Figure 2a*). In WT U937 cells, the activity of the *IL6* promoter was increased after lipopolysaccharide (LPS) treatment, and *FAM76B* knockout in U937 cells made this change even more prominent (*Figure 2b*). Next, to evaluate if FAM76B regulates *IL6* promoter activity via influencing NF-κB, we constructed luciferase reporters controlled by a miniCMV promoter containing the NF-κB binding motif 1 or 2 sequence from the *IL6* promoter region. The results indicated that the luciferase activity from the vector in both WT and *Fam76b-/-* U937 cells was increased, especially in the latter, after treatment with 1 ng/mL LPS (*Figure 2c and d*). These data suggested that FAM76B inhibited the activity of the *IL6* promoter by downregulating the NF-κB pathway.

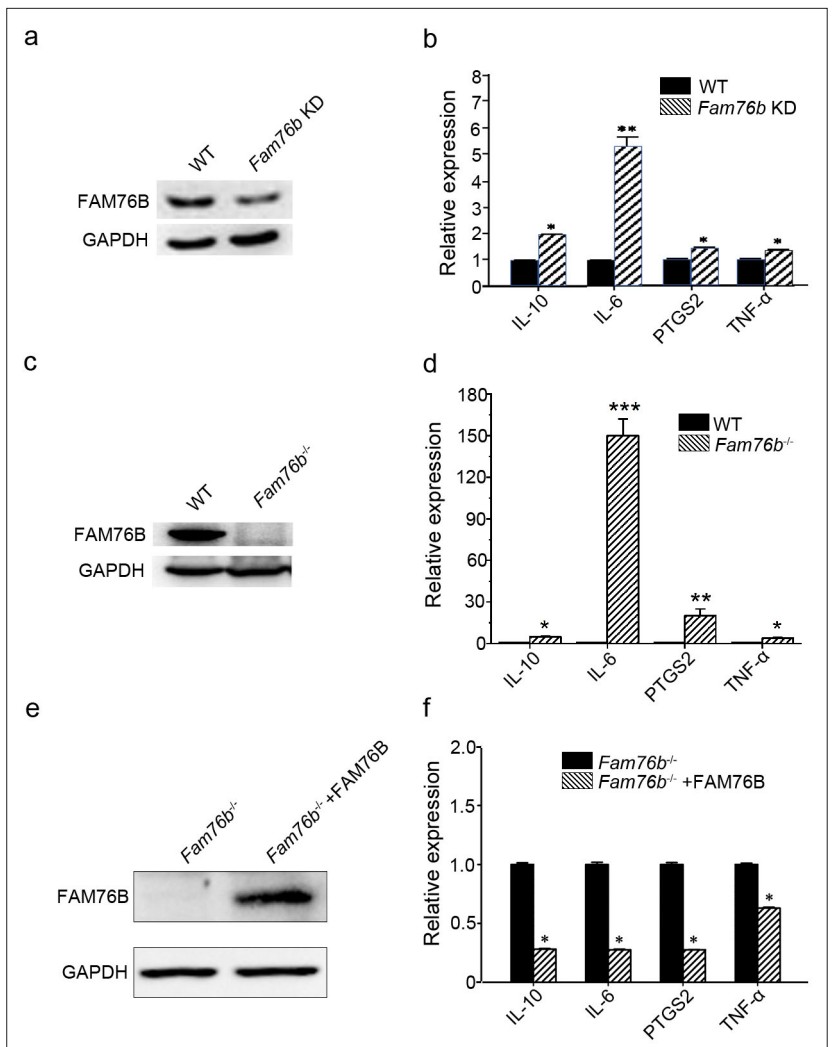

**Figure 1.** FAM76B regulates the expression of cytokines in U937 cells. (**a**) Western blot for the detection of FAM76B expression in U937 with *FAM76B* knockdown cells (*Fam76b* KD) generated with the Cas9/sgRNA technique. (**b**) *FAM76B* knockdown in U937 increased the expression of *IL6*, *PTGS2*, *TNFα* and *IL10*, as determined by real-time PCR in the presence of PMA+LPS/IFNγ. (**c**) Western blot for the detection of FAM76B expression in U937 with *FAM76B* knockout cell line (*Fam76b*^{-/-}) generated with the Cas9/sgRNA technique. (**d**) *FAM76B* knockout in U937 cells significantly enhanced the expression of *IL6*, *PTGS2*, *TNFα* and *IL10*, as determined by real-time PCR in the presence of PMA+LPS/IFNγ. (**e**) Western blot validated the rescued expression of FAM76B in the *Fam76b*^{-/-} U937 cell line infected with the lentiviral vector expressing FAM76B. (**f**) The rescued expression of FAM76B in the U937 cell line with *FAM76B* knockout reduced the mRNA levels of cytokines in the presence of PMA +LPS/IFNγ. The experiments were performed at least three times. Values are mean ± SD; *p<0.05, **p<0.01, ***p<0.001, statistically significant.

The online version of this article includes the following source data and figure supplement(s) for figure 1:

**Source data 1.** Labeled uncropped western blot images for *Figure 1*.

**Source data 2.** Raw western blot images with tiff format source data for *Figure 1*.

**Figure supplement 1.** Confirmation by sequencing of Cas9/sgRNA-mediated *FAM76B* knockout in the U937 cell line.

To further explore the mechanisms of FAM76B regulating inflammation via NF-κB, the FAM76B-interacting proteins were investigated using immunoprecipitation coupled to mass spectrometry (IP-MS) in U937 cells. We identified 160 proteins that interacted with FAM76B within U937 cells. Selective molecules of these proteins were listed in *Supplementary file 1*, with their interaction scores and ranks. To validate the IP-MS results, co-immunoprecipitation on selected proteins was performed.

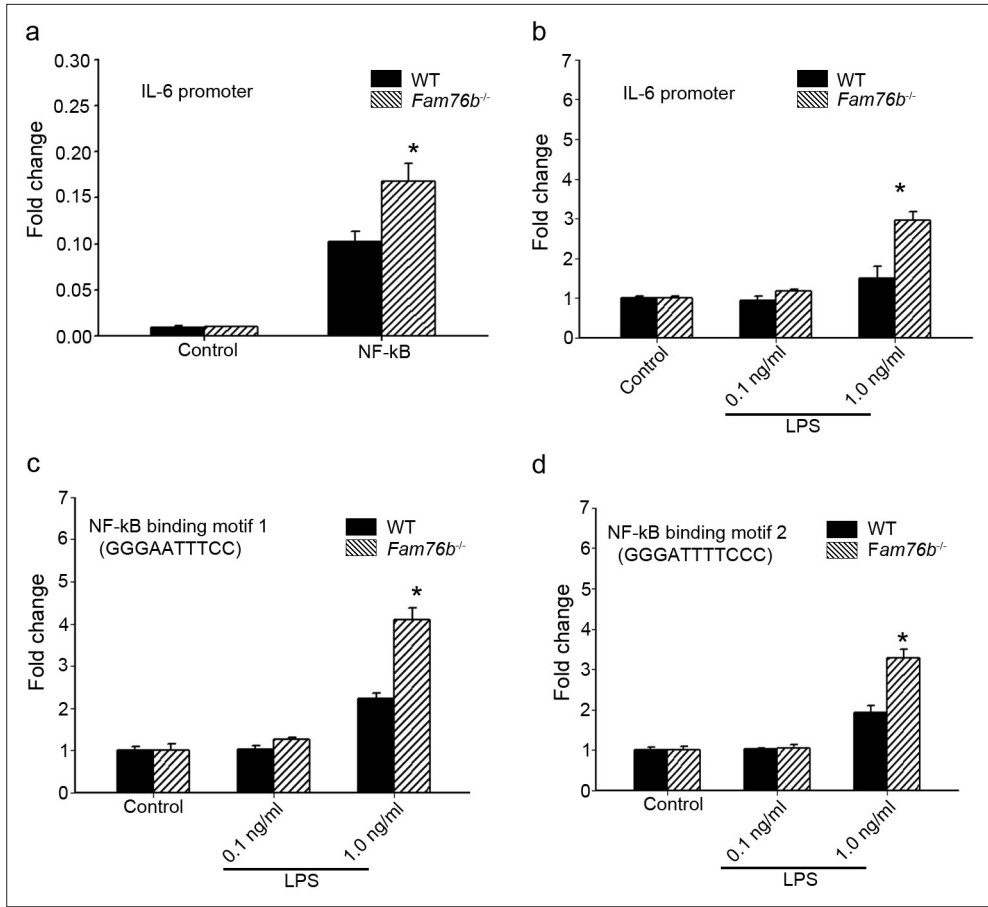

**Figure 2.** FAM76B regulates *IL6* promoter activity by affecting the NF-$\kappa$B pathway. (**a**) The luciferase reporter vector of the *IL6* promoter was tested in wild-type (WT) and *FAM76B* knockout (*Fam76b*[-/-]) HEK293 cells. The luciferase activity was low in both WT and *FAM76B* knockout HEK293 cells and significantly increased after transfecting the NF-$\kappa$B-expressing vector. The increased activity of the *IL6* promoter was more prominent in *Fam76b*[-/-] HEK293 cells than in WT cells. (**b**) The *IL6* promoter was increased in WT and *Fam76b*[-/-] U937 cells, but was more prominent in the latter, after lipopolysaccharide (LPS) treatment. (**c and d**) Luciferase activity was increased in WT and *Fam76b*[-/-] U937 cells carrying NF-$\kappa$B binding motifs 1 or 2 (and more prominently in the *Fam76b*[-/-] U937 cells), indicating that FAM76B inhibited NF-$\kappa$B binding activity of *IL6* promoter. The experiments were repeated at least three times. Values are mean ± SD; *p<0.05, statistically significant.

Among those FAM76B interacting proteins, hnRNPs captured our attention because of their reported function in regulating the NF-κB pathway (***Zhao et al., 2009***; ***Ma et al., 2022***). The hnRNPA2B1 was selected for further validation. The co-immunoprecipitation results confirmed the interaction of FAM76B with hnRNPA2B1 (***Figure 3a***). In addition, confocal microscopy also revealed the co-localization of FAM76B and hnRNPA2B1 in the nucleus of HEK293 cells overexpressing FAM76B-eGFP and hnRNPA2B1-mCherry (***Figure 3b***). Based on the validation of the interaction between hnRNPA2B1 and FAM76B, to identify the domain(s) of the hnRNPA2B1 required for their interaction with FAM76B, we generated different truncates containing RRM1, RRM2, or RGD domains of hnRNPA2B1 and demonstrated that the RGD domain of hnRNPA2B1 was responsible for its binding to FAM76B (***Figure 3c and d***). It has been reported that hnRNPA1 binds to I$\mathrm{K}$B$\alpha$, which led to IκBα degradation and consequently, to NF-κB activation (***Zhao et al., 2009***). Using co-immunoprecipitation with HEK293 cells transfected with plasmids expressing hnRNPA2B1 and IκBα-flag (or IκBε-flag), we also demonstrated that hnRNPA2B1 bound to IκBα or IκBε (***Figure 3e***). Based on the results above, we speculated that FAM76B, hnRNP A/B, and IκBs could form a protein complex by binding FAM76B to hnRNPs through the RGD domain and binding hnRNPs to IκBs through the RRM2 domain (***Figure 3f***). These results suggested that FAM76B might regulate hnRNPA2B1-mediated inflammatory action through a novel interaction, subsequently influencing the activation of NF-kB pathway.

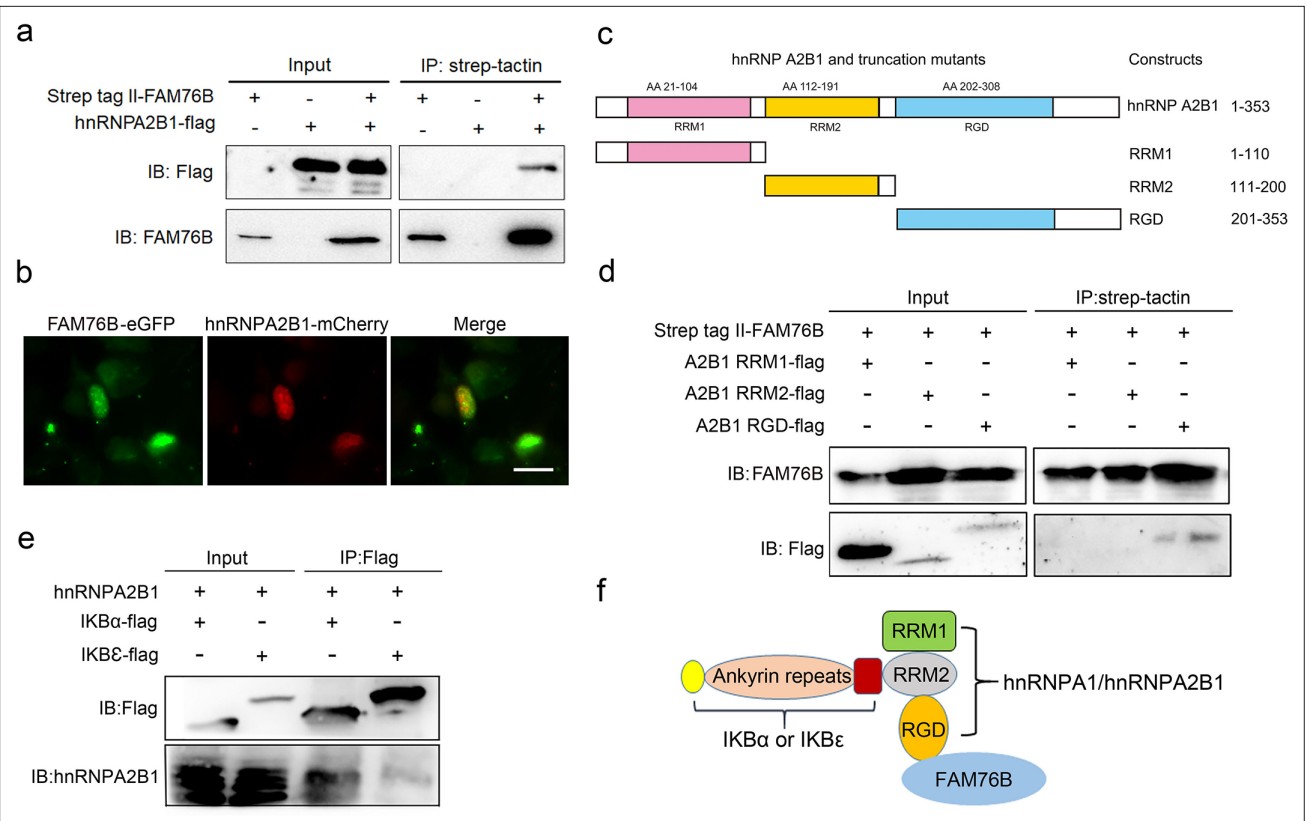

**Figure 3.** Validation of the interactions among FAM76B, hnRNPA2B1 and I κ Bs. (**a**) The interaction between FAM76B and hnRNPA2B1 was revealed by co-immunoprecipitation. FAM76B-Strep-tag II and hnRNPA2B1-Flag were overexpressed in HEK293 cells, followed by co-immunoprecipitation of FAM76B and hnRNPA2B1 using whole cell lysate and western blot with anti-Flag or anti-FAM76B antibodies. (**b**) Confocal microcopy revealed the co-localization of FAM76B and hnRNPA2B1 in the nucleus of HEK293 cells transfected with plasmids expressing FAM76B-eGFP and hnRNPA2B1-mCherry, respectively. Scale bar, 20 μm. (**c**) An illustration of hnRNPA2B1 domains (RRM1, RRM2, and RGD) tagged with Flag generated for detecting the hnRNPA2B1 region(s) responsible for binding FAM76B. (**d**) Identification of the hnRNPA2B1 domains responsible for binding to FAM76B. Strep tag II-FAM76B and hnRNPA2/B1 domain-flag were transfected into HEK293 cells, followed by co-immunoprecipitation and western blot with anti-Flag or anti-FAM76B antibodies, which showed the interaction between the RGD domain of hnRNPA2B1 and FAM76B. (**e**) Similarly, the interaction between hnRNPA2B1 and I κ Bα-flag or I κ Bε-flag was detected by co-immunoprecipitation. (**f**) Schematic diagram of FAM76B, hnRNPA2B1 and I κ B s' protein complex formation: hnRNPA2B1 binds FAM76B by its RGD domain and binds I κ Bs by its RRM2 domain.

The online version of this article includes the following source data for figure 3:

**Source data 1.** Labeled uncropped western blot images for *Figure 3*.

**Source data 2.** Raw western blot images with tiff format for *Figure 3*.

FAM76B and hnRNPA2/B1 are nuclear-localized proteins; however, it has been reported that hnRNPA2/B1 could translocate from the nucleus to the cytoplasm, which would lead to activation of the NF-κB pathway (*Wang et al., 2019*). Considering the interaction of FAM76B and hnRNPA2B1, we speculated that FAM76B was the molecule that affected the cytoplasmic translocation of hnRNPA2B1 and subsequent NF-kB activation and inflammation. To test that hypothesis, hnRNPA2B1 immunostaining was performed on U937 cells with *FAM76B* knockout treated with PMA, which showed increased cytoplasmic translocation of hnRNPA2B1 (*Figure 4a*); this result was also confirmed by the levels of nuclear and cytoplasmic hnRNPA2B1 using western blot and semiquantification based on the results of western blot (*Figure 4b and c*). A similar result was obtained in U937 cells with *FAM76B* knockout treated with PMA followed by LPS+IFNγ stimulation (*Figure 4d–f*). To further confirm our hypothesis, we produced stable U937 cell lines overexpressing FAM76B (*Fam76b^OE*) (*Figure 4—figure supplement 1a*). The inflammatory responses in this stable cell line were examined following treated with PMA plus LPS+IFNγ stimulation. Overexpression of FAM76B downregulated *IL6* expression significantly, and to a lesser extent, the expressions of *PTGS2* and *TNFα* (*Figure 4—figure supplement 1b*). *IL10* expression was minimally or mildly affected by FAM76B overexpression

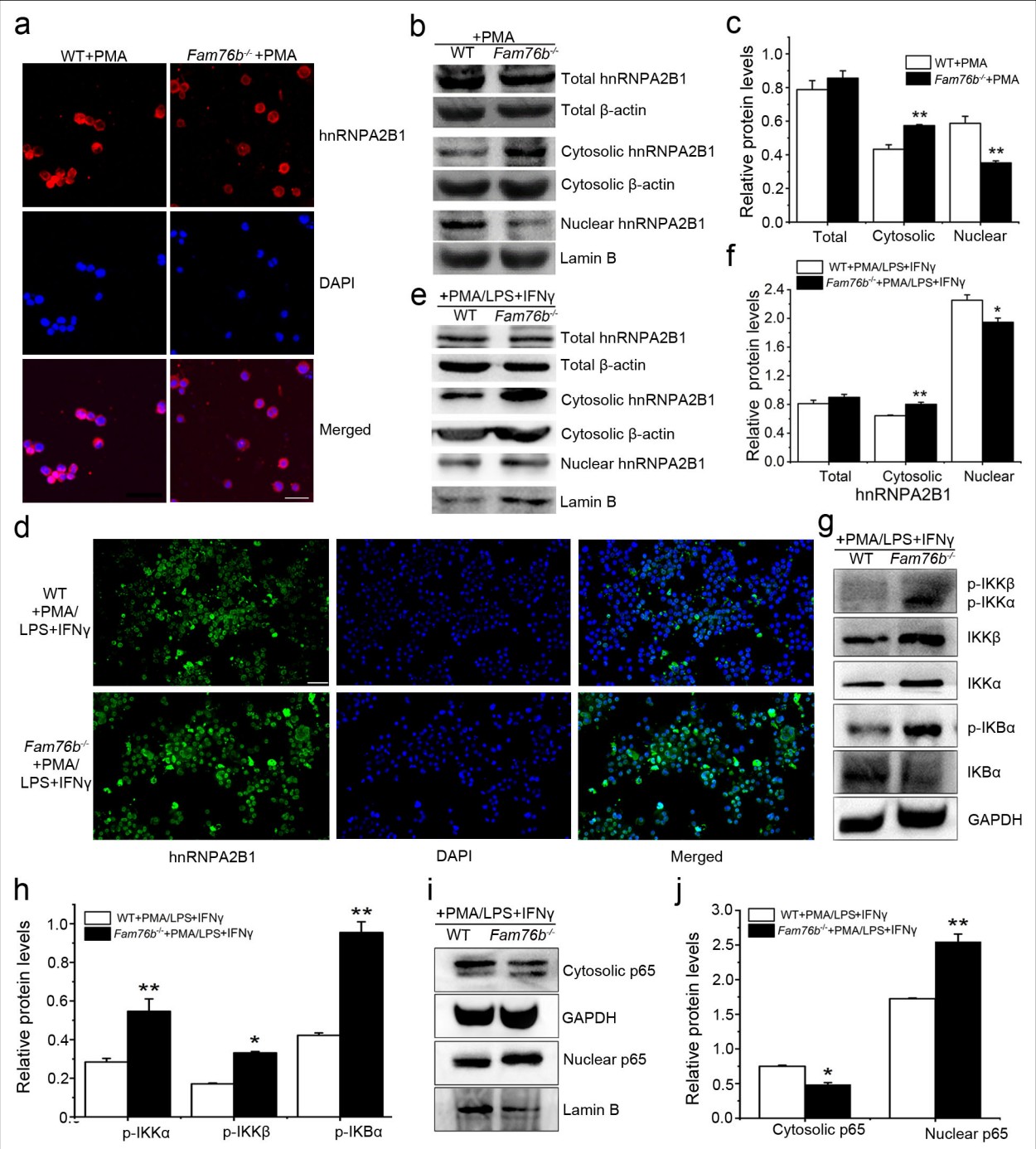

**Figure 4.** FAM76B regulates the NF-$\kappa$B pathway via influencing the translocation of hnRNPA2B1. (**a**) Immunofluorescence revealed increased cytoplasmic translocation of hnRNPA2B1 in U937 cells with *FAM76B* knockout (*Fam76b$^{-/-}$*) stimulated with PMA for 48 hr. Scale bar, 20 μm. (**b**) Western blot confirmed the cytoplasmic translocation of hnRNPA2B1 in *Fam76b$^{-/-}$* U937 cells stimulated with PMA for 48 hr. (**c**) The semiquantification of the results of western blot from (b). (**d**) Immunofluorescence revealed increased cytoplasmic translocation of hnRNPA2B1 in *Fam76b$^{-/-}$* U937 cells stimulated with PMA followed by LPS+IFNγ. Scale bar, 50 μm. (**e**) Western blot confirmed the cytoplasmic translocation of hnRNPA2B1 in *Fam76b$^{-/-}$* U937 cells stimulated with PMA followed by LPS+IFNγ. (**f**) The semiquantification of the results of western blot from (e). (**g**) Western blot revealed the increased phosphorylation of endogenous IKKα, IKKβ and I$\kappa$Bα in *Fam76b$^{-/-}$* U937 cells with 1 ng/mL PMA followed by LPS+IFNγ treatment. (**h**) The semiquantification of the western blots result from (g). (**i**) Western blot showed the increased nuclear translocation of p65 in *Fam76b$^{-/-}$* U937 cells stimulated with PMA followed by incubation with LPS+IFNγ. (**j**) The semiquantification of the western blot result from (i). The experiments were performed at least three times. Values are mean ± SD; *p<0.05, **p<0.01, statistically significant.

The online version of this article includes the following source data and figure supplement(s) for figure 4:

*Figure 4 continued on next page*

*Figure 4 continued*

**Source data 1.** Labeled uncropped western blot images for *Figure 4*.

**Source data 2.** Raw western blot images with tiff format for *Figure 3*.

**Figure supplement 1.** The downregulation of cytokines in FAM76B overexpressed U937 cells treated with PMA followed by LPS+IFNγ.

**Figure supplement 1—source data 1.** Labeled uncropped western blot images for *Figure 4—figure supplement 1*.

**Figure supplement 1—source data 2.** Raw western blot images with tiff format for *Figure 4—figure supplement 1*.

**Figure supplement 2.** No significant change in cytoplasmic translocation of hnRNPA2B1 in FAM76B overexpressed U937 cells treated with PMA only.

**Figure supplement 2—source data 1.** Labeled uncropped western blot images for *Figure 4—figure supplement 2*.

**Figure supplement 2—source data 2.** Raw western blot images with tiff format for *Figure 4—figure supplement 2*.

**Figure supplement 3.** Decreased cytoplasmic translocation of hnRNPA2B1 in FAM76B overexpressed U937 cells treated with PMA followed by LPS+IFNγ.

**Figure supplement 3—source data 1.** Labeled uncropped western blot images for *Figure 4—figure supplement 3*.

**Figure supplement 3—source data 2.** Raw western blot images with tiff format for *Figure 4—figure supplement 3*.

**Figure supplement 4.** Increased cytoplasmic translocation of hnRNPA2B1 and FAM76B in U937 cells treated with PMA followed by LPS+IFNγ.

in U937 cells (*Figure 4—figure supplement 1b*). These data indicated that FAM76B had an anti-inflammatory effect. At the same time, there were no significant differences in cytoplasmic translocation of hnRNPA2B1 between U937 cells and U937 cells overexpressing FAM76B, when they were treated with PMA only (*Figure 4—figure supplement 2*). However, the decreased cytoplasmic translocation of hnRNPA2B1 was found in $Fam76b^{OE}$ U937 cells when were treated with PMA followed by LPS+IFNγ (*Figure 4—figure supplement 3*). Furthermore, we found that *FAM76B* knockout in U937 cells resulted in increased phosphorylation of endogenous IKKα, IKKβ and the downstream molecule IκBα upon LPS stimulus (*Figure 4g*), which was also confirmed by semiquantification (*Figure 4h*). In addition, we observed a concurrent increase in the nuclear translocation of p65, a hallmark of classical NF-κB pathway activation, by western blot and semiquantification in $Fam76b^{-/-}$ U937 cells by PMA plus LPS+IFNγ stimulation (*Figure 4i and j*). The results above indicated that FAM76B could inhibit NF-κB activation by affecting the translocation of hnRNPA2B1. Interestingly, when WT U937 cells was induced into the M1-like macrophage state with PMA followed by LPS+IFNγ stimulation, both FAM76B and hnRNPA2B1 were found to be partially translocated into the cytoplasm from the nucleus (*Figure 4—figure supplement 4*). The possible role of increased FAM76B in the cytoplasm needs to be further studied.

## Inflammation mediated by macrophages and microglia are enhanced in *Fam76b* knockout C57BL/6 mice

Our previous study showed that FAM76B is widely expressed in different human organs, with the highest expression levels found in the brain and spleen (*Zheng et al., 2016*). Similarly, here we found that the mouse brain and spleen also had high levels of FAM76B expression by real-time PCR (*Figure 5—figure supplement 1*) which suggested that FAM76B might have important functions related to inflammation in these tissues. To investigate whether FAM76B had anti-inflammatory activity in vivo, we produced *Fam76b* knockout C57BL/6 mice ($Fam76b^{-/-}$ mice) by gene trap technology (*Figure 5a*). *FAM76B* knockout was confirmed by genotyping analysis and the expression of *FAM76B* at the mRNA and protein level (*Figure 5b–d*). $Fam76b^{-/-}$ mice showed normal weight gain and lifespan (*Figure 5—figure supplement 2*). $Fam76b^{-/-}$ mice developed spleens that were visibly enlarged and weighed more than those of WT mice (*Figure 6a and b*). Hematoxylin and eosin (H&E)-stained sections of the spleen showed normal red pulp and hypertrophy of white pulp (*Figure 6a*). Flow cytometry revealed an increase in the CD11b+ myeloid population and a slight decrease in CD19+ B cells in $Fam76b^{-/-}$ mice, while no changes were found in CD3+ T cells (*Figure 6c–g*). To further elucidate the effect of FAM76B on the function of macrophages in the $Fam76b^{-/-}$ mice, we examined their spleens following intraperitoneal LPS injection. There were no significant morphologic changes in the WT and $Fam76b^{-/-}$ mice without treatments (*Figure 6h*). However, after LPS treatment, the $Fam76b^{-/-}$ mice had many tingible body macrophages in the white pulp of the spleens, while the WT mice had none (*Figure 6h*). These data suggested that loss of FAM76B expression might lead to a decreased ability of the white pulp macrophages to clear out the apoptotic cells produced during the

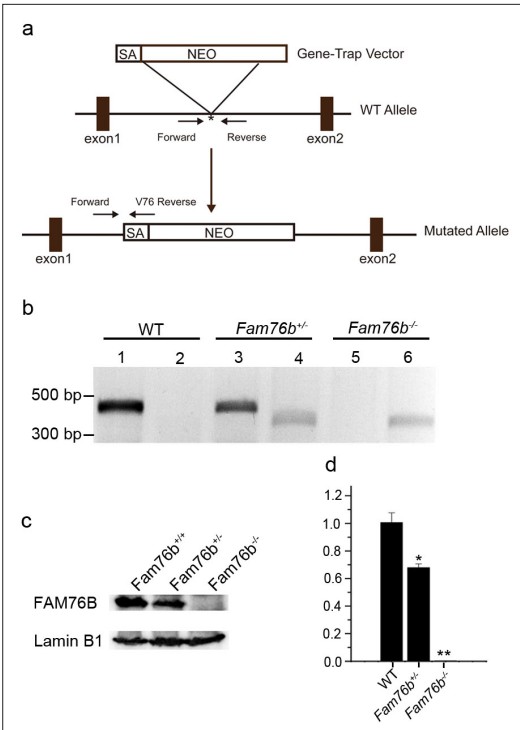

**Figure 5.** Generation of *Fam76b* knockout (*Fam76b⁻/⁻*) mice. (**a**) Schematic diagram of the homologous recombination construct for generating *Fam76b⁻/⁻* mice, with arrows denoting primer locations. (**b**) Genotyping results for *Fam76b* wild-type (WT), hemizygous (+/-), and homozygous (-/-) mice. Lanes 1, 3, and 5 are the amplified products using *Fam76b* forward and reverse primers (***Supplementary file 3***); lanes 2, 4, and 6 are the amplified products using *Fam76b* forward and V76 reverse primers (***Supplementary file 3***). (**c**) FAM76B protein levels in mouse embryonic fibroblasts from *Fam76b* WT, hemizygous (+/-), and homozygous (-/-) mice were confirmed by western blot (n=3, each group). (**d**) Detection of *Fam76b* mRNA levels in mouse embryonic fibroblasts from *Fam76b* WT, hemizygous (+/-), and homozygous (-/-) mice by real-time PCR (n=3, each group). Values are mean ± SD; *p<0.05, **p<0.01, statistically significant.

The online version of this article includes the following source data and figure supplement(s) for figure 5:

**Source data 1.** Labeled uncropped western blot images for *Figure 5*.

**Source data 2.** Raw western blot images with tiff format for *Figure 5*.

**Figure supplement 1.** The detection of *FAM76B* expression in mouse tissues by real-time PCR (n=5).

**Figure supplement 2.** Characterization of *Fam76b* knockout mice.

germinal center reaction. The results above indicated that FAM76B might be involved in macrophage function.

Macrophages play an important role in inflammation. So to further investigate the effect of FAM76B on the regulation of macrophage-mediated inflammation in vivo, bone marrow-derived macrophages (BMDMs) from *Fam76b⁻/⁻* mice were isolated and used to analyze *IL6* expression. The results indicated that macrophages from *Fam76b⁻/⁻* mice could significantly increase *IL6* expression compared to WT murine macrophages (***Figure 6—figure supplement 1a***). Moreover, the increased expression of *IL6* could be downregulated when FAM76B was rescued in macrophages from *Fam76b⁻/⁻* mice (***Figure 6—figure supplement 1b***), which was consistent with the results obtained from *Fam76b⁻/⁻* U937 cells in vitro. The results above indicated that FAM76B possessed anti-inflammation activity in vivo.

Microglia play a crucial role in mediating neuroinflammation. We found that FAM76B and IBA-1, a marker of microglia activation, were co-localized in the mice brain tissue (***Figure 7—figure supplement 1***). Therefore, we tested whether deleting FAM76B could enhance inflammation mediated by microglia in *Fam76b⁻/⁻* mice. First, immunostaining with anti-IBA-1 antibody (***Ito et al., 1998***; ***Okere and Kaba, 2000***; ***Hirayama et al., 2001***) was used to reveal the total microglial population in the hippocampus and thalamus of *Fam76b⁻/⁻* mice. IBA1-positive microglia were higher in 12-month-old *Fam76b⁻/⁻* mice than that in age-matched WT mice, and displayed a highly reactive morphologies with abundant cytoplasm (***Figure 7a***). However, there were no significant differences between 4-month-old *Fam76b⁻/⁻* and WT mice (data not shown). An evaluation of the densities of IBA-1-positive microglia (number/mm²) showed more IBA-1-positive microglial cells in the CA1 region and thalamus of 12-month-old *Fam76b⁻/⁻* mice than in those same brain areas of age-matched WT mice, indicating an enhanced activation of microglia and neuroinflammation. The densities of IBA-1-positive microglia were similar between 4-month-old *Fam76b⁻/⁻* and WT mice (***Figure 7b***). A controlled cortical impact (CCI) mouse model (***Zheng et al., 2022***), as a widely used model for TBI, was then used to examine the effect of FAM76B on microglia-mediated neuroinflammation. The densities of IBA-1-positive microglia were higher in the hippocampus adjacent to the cortical contusion of *Fam76b⁻/⁻* mice than in that of WT mice or sham controls (***Figure 7c and d***). Real-time PCR showed that the expression level of *IL6* in the ipsilateral

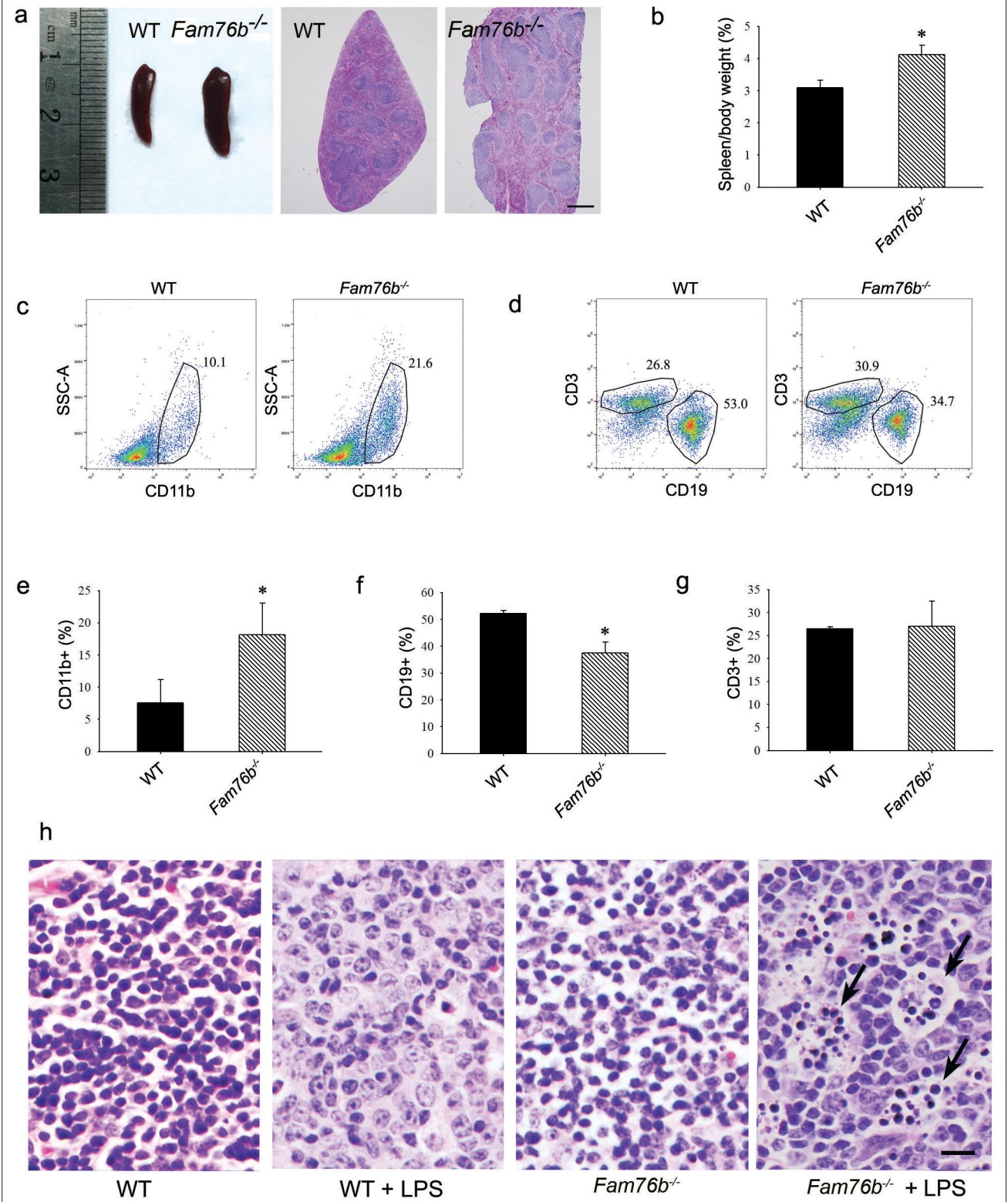

**Figure 6.** *Fam76b* knockout mice have enlarged spleens with altered cell populations and inflammation. (**a**) Enlarged spleen with white pulp hypertrophy in *Fam76b* knockout mice (*Fam76b*⁻/⁻ mice) (5 months), as compared with wild-type (WT). Scale bar, 500 μm. (**b**) The weight ratio of spleen to body in *Fam76b*⁻/⁻ mice (5 months) revealed an enlarged spleen, as compared with WT. *p<0.05, statistically significant (n=5, each group). (c–g) Flow cytometry results of the cell population of spleens of *Fam76b*⁻/⁻ mice (5 months). Dot plots (**c and d**) and bar graphs (e, f, and g) show increased populations of CD11b+ and decreased populations of CD19+ B cells in *Fam76b*⁻/⁻ mice spleens (n=4, each group) ( Values are mean ± SD;

*Figure 6 continued on next page*

*Figure 6 continued*

*p<0.05, one-way ANOVA. (**h**) Photomicrographs of hematoxylin and eosin (H&E)-stained spleens from *Fam76b-/-* mice intraperitoneally injected with lipopolysaccharide (LPS) showed abundant tingible body macrophages (arrows) in the germinal center, whereas no tingible body macrophages were seen in LPS-treated WT or in phosphate-buffered saline (PBS)-treated *Fam76b-/-* and WT mice (n=5, each group). Scale bar, 25 μm.

The online version of this article includes the following figure supplement(s) for figure 6:

**Figure supplement 1.** Lipopolysaccharide (LPS) stimulated bone marrow-derived macrophages (BMDMs) from *Fam76b* knockout mice showed increased IL6 expression.

hippocampus was elevated in both *Fam76b-/-* and WT mice 3 days after TBI, with more prominent changes observed in *Fam76b-/-* mice (**Figure 7e**). The above results indicated that, in the mouse experimental TBI model by CCI, the loss of FAM76B could induce microglia activation and promote neuroinflammation in mouse brain after TBI. We then stereotactically injected LPS into the left frontal lobes of *Fam76b-/-* and WT mice. Seven days post-treatment, histologic examination revealed abundant macrophages at the injection site of *Fam76b-/-* mice, but only a mild inflammatory response in WT mice (**Figure 7—figure supplement 2a**). The results of real-time PCR and enzyme-linked immunosorbent assay (ELISA) showed that the expression level of IL-6 was elevated in both WT and *Fam76b-/-* mice 24 hr post-LPS treatment, with more prominent changes evident in the *Fam76b-/-* mice (**Figure 7— figure supplement 2b and d**). Tumor necrosis factor alpha (TNF-α) expression was similarly altered as in the WT and *Fam76b-/-* mice, but to a lesser extent than IL-6 (**Figure 7—figure supplement 2c and e**). The above data indicated that after the occurrence of neuroinflammation (LPS-injected) or brain injection injury (phosphate buffered saline [PBS]-injected), knockout of *FAM76B* could lead to further enhancement of the inflammatory response. Overall, considering that the neuroinflammation was an important pathological feature of TBI, our data in mouse model, including the experimental TBI model by CCI and the intracortical LPS injection model, supported that FAM76B played an important role in TBI by regulating the activation of microglia and inflammatory response after TBI and indicated that FAM76B was also involved in modulating neuroinflammation mediated by microglial cells in vivo.

## Co-localization of FAM76B and hnRNPA2B1 in the cytoplasm is increased in the brain of the patients with TBI or neurodegeneration

Neuroinflammation is known to be closely related the development of TBI and neurodegeneration. Our previous study showed that FAM76B was expressed higher in the brain and spleen than other human organs (***Zheng et al., 2016***). To assess whether FAM76B was involved in the neuroinflammation associated with the human diseases named above, we compared the distribution and expression of FAM76B in diseased brain tissues to those of normal brains. In the normal brain, immunohistochemical stains revealed weak cytoplasmic staining of FAM76B in neurons and nuclear staining in glial cells (**Figure 8—figure supplement 1a and b**). In areas of organizing necrosis in brains with TBI, we found that macrophages—which were labeled by the microglial/macrophage marker IBA-1 (***Ito et al., 1998***; ***Okere and Kaba, 2000***; ***Hirayama et al., 2001***)—were all strongly immunopositive for FAM76B (**Figure 8—figure supplement 1c**). In the hippocampal CA1 area of brains with acute ischemic injury, FAM76B immunostains highlighted many reactive microglial cells (**Figure 8—figure supplement 1d**). These findings strongly suggested that microglial FAM76B was upregulated in injured brains and thus might have important functions in regulating neuroinflammation.

TBI elicits neuroinflammation, which is essential for proper tissue regeneration and recovery (***Simon et al., 2017***). Thus, we next studied the role of FAM76B in TBI-induced neuroinflammation by examining the expression and cytological distribution of FAM76B/hnRNPA2B1 in human brains with TBI. In normal brain tissue, FAM76B was mainly localized in the nuclei of glial cells, including microglial (**Figure 8a**) and oligodendroglial (**Figure 8—figure supplement 1b**) cells. HnRNPA2B1 was found to be co-localized with FAM76B in the nuclei of these cells (**Figure 8b**). In acute and organizing TBI, in response to the contusion, there was increased expression and cytoplasmic translocation of microglial FAM76B (**Figure 8a**), which was co-localized with hnRNPA2B1 (**Figure 8b**). In chronic TBI, the residual macrophages/microglia in the previously injured area showed a persistent cytoplasmic FAM76B/hnRNPA2B1 distribution; moreover, the FAM76B- and hnRNPA2B1-positive microglia appeared dystrophic in morphology (**Figure 8a and b**). These results were consistent with those

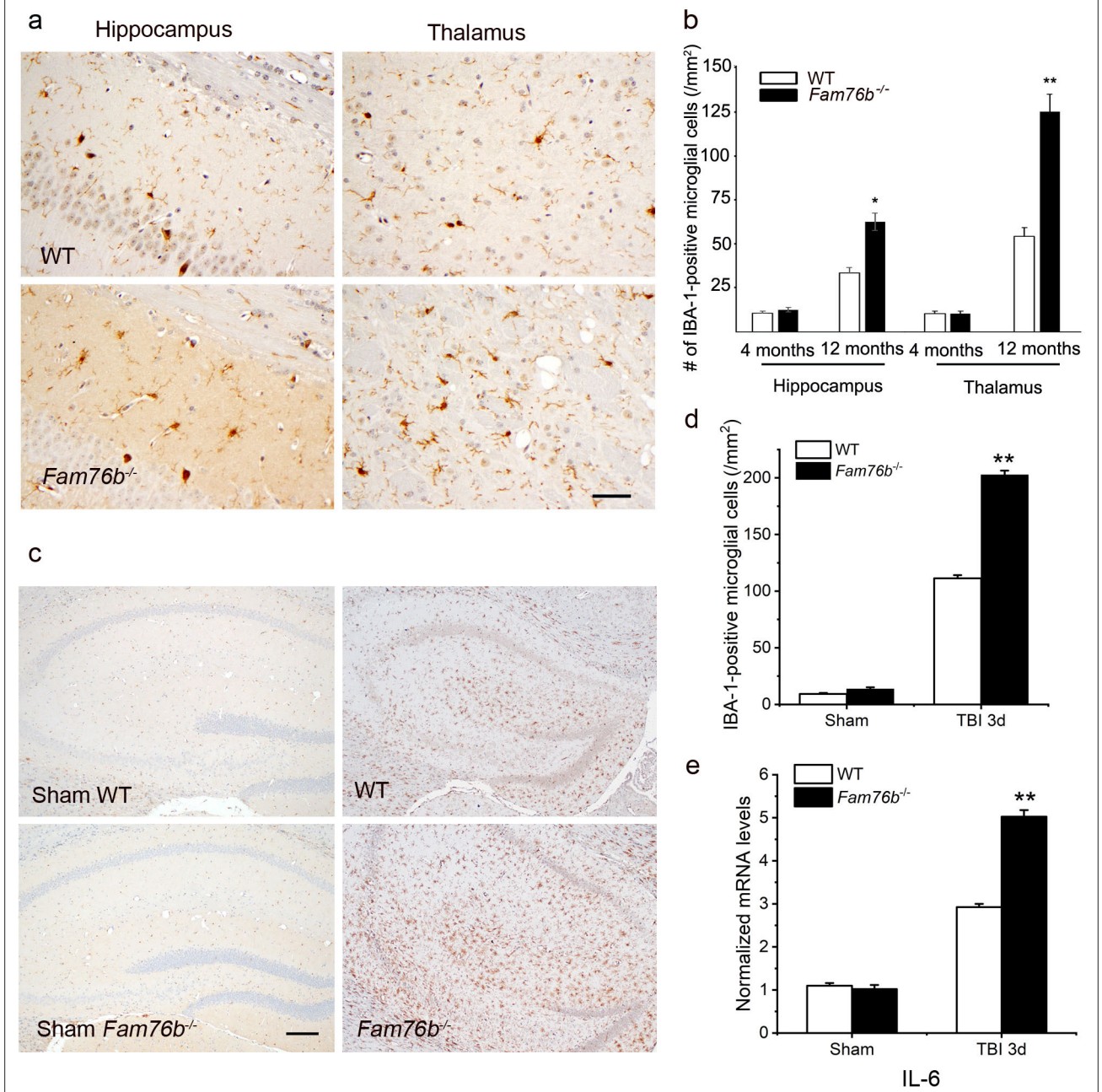

**Figure 7.** Neuroinflammation is enhanced in *Fam76b* knockout mice. (**a**) Increased IBA-1-positive microglial infiltration in the hippocampus (lower left panel) and thalamus (lower right panel) of 12-month-old *Fam76b* knockout mice (*Fam76b⁻/⁻* mice), as compared with age-matched wild-type (WT) mice (n=5, each group). Scale bar, 50 μm. (**b**) Density of IBA-1-positive microglia in hippocampal CA1 regions and the thalamus of 4- and 12-month-old *Fam76b⁻/⁻* mice, as compared with age-matched WT mice (n=5, each group). (**c**) Increased IBA-1-positive microglia in the hippocampus adjacent to the contusion site of *Fam76b⁻/⁻* mice, as compared with WT mice and sham controls. Scale bar, 200 μm. (**d**) Density of IBA-1-positive microglia in hippocampal CA1 regions of *Fam76b⁻/⁻* mice, as compared with WT mice and sham controls (n=5, each group). Values are mean ± SD. The densities of IBA-1-positive microglia were compared to the control value by Student's t-test (**p<0.01). (**e**) Increased *IL6* expression, as revealed by real-time PCR, in the ipsilateral hippocampus in both *Fam76b⁻/⁻* and WT mice 3 days after traumatic brain injury (TBI), with more prominent changes in *Fam76b⁻/⁻* mice. Values are mean ± SD; *p<0.05, **p<0.01, statistically significant.

The online version of this article includes the following figure supplement(s) for figure 7:

**Figure supplement 1.** The co-localization of FAM76B with IBA-1 in the mouse brain by immunofluorescence staining.

**Figure supplement 2.** Neuroinflammation in *Fam76b* knockout mice after lipopolysaccharide (LPS) treatment.

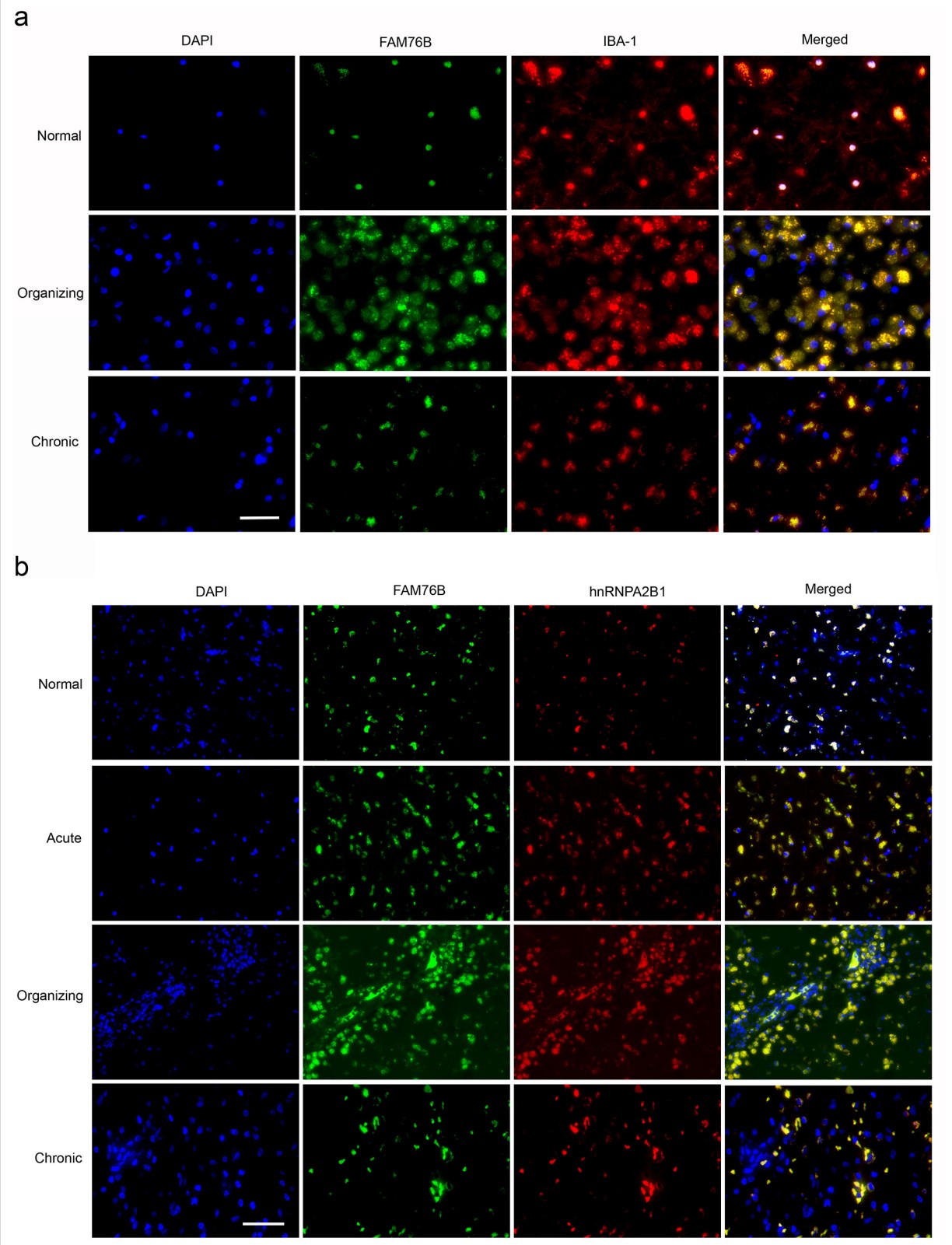

**Figure 8.** The expression and distribution of FAM76B/hnRNPA2B1 in human brains with traumatic brain injury (TBI). (**a**) Microglial localization of FAM76B in organizing and chronic TBI. Immunofluorescence revealed the nuclear localization of FAM76B in IBA-1-positive microglia in the normal human cortex. FAM76B was upregulated and cytoplasmically translocated in microglia/macrophages in the human cortex with organizing TBI. In chronic TBI, microglial FAM76B showed persistent cytoplasmic distribution. Scale bar, 100 μm. (**b**) Upregulation and cytoplasmic translocation of FAM76B and hnRNPA2B1 in

*Figure 8 continued on next page*

*Figure 8 continued*
acute, organizing, and chronic TBI. Immunofluorescence revealed the nuclear co-localization of FAM76B and hnRNPA2B1 in the normal human cortex. Both FAM76B and hnRNPA2B1 were upregulated and cytoplasmically translocated in microglia/macrophages in the cortex of a patient with acute TBI. Both proteins were further upregulated in the cytoplasm of the microglia/macrophages in the human cortex with organizing TBI. In chronic TBI, microglial FAM76B and hnRNPA2B1 showed persistent cytoplasmic distribution and co-localization. Scale bar, 100 μm.

The online version of this article includes the following figure supplement(s) for figure 8:

**Figure supplement 1.** Expression of FAM76B in normal and diseased tissues.

obtained in U937 cells, supporting the conclusion that FAM76B affects hnRNPA2B1's translocation from the nucleus to the cytoplasm.

Neuroinflammation is an important contributor to neurodegeneration. Hence, we evaluated the role of FAM76B in the common neurodegenerative diseases Alzheimer's disease (AD), frontotemporal lobar degeneration with tau pathology (FTLD-tau), and frontotemporal lobar degeneration with TAR DNA-binding protein 43 inclusions (FTLD-TDP). Similar to what we observed in chronic TBI, FAM76B and hnRNPA2B1 were co-localized in the microglial cytoplasm of brain tissue from patients with neurodegeneration, and the FAM76B- and hnRNPA2B1-positive microglia appeared dystrophic in morphology (*Figure 9a*). In addition, immunostains revealed that IBA-1- and FAM76B-positive microglia were scattered in the frontal cortex of normal aging controls and were slightly more numerous in that of AD, FTLD-tau, and FTLD-TDP patients (*Figure 9b*). Moreover, the cellular densities of FAM76B-positive and IBA-1-positive microglia were both higher in AD, FTLD-tau, and FTLD-TDP patients than in normal aging controls (IBA-1-positive microglia, **$p<0.01$; FAM76B-positive microglia, ##$p<0.01$) (*Figure 9c*). The ratio of FAM76B-positive microglial density to IBA-1-positive total microglia density was the highest in the cortex in FTLD-TDP patients (&&$p<0.01$) (*Figure 9c*). These data suggested that the FAM76B regulated the inflammatory process in AD, FTLD-tau, and FTLD-TDP brains, but is most prominent in FTLD-TDP.

## Discussion

The key findings of this study are as follows. (a) FAM76B was a novel systemic inflammatory and neuroinflammatory modulator that could inhibit NF-κB activity by affecting the cytoplasmic translocation of hnRNPA2B1, a newly discovered interacting partner of FAM76B in this study. (b) Human brains with TBI showed the activation, evolution, and persistence of FAM76B-positive microglia, indicating the significant role of FAM76B in the development of human TBI. (c) There was an increased cytoplasmic co-localization of FAM76B and hnRNPA2B1 in the brain of the patients with TBI or neurodegeneration, particularly in FTLD-TDP.

This is the first report investigating the function of FAM76B in inflammation and neuroinflammation. In this study, we speculated that when FAM76B was present in the nucleus, the hnRNPA2B1 protein was trapped in the nucleus by its binding to FAM76B; however, when FAM76B was decreased or made absent (such as by *FAM76B* knockdown or knockout, respectively) or was subjected to inflammatory stimulation (such as by LPS), the hnRNPA2B1 protein translocated to the cytoplasm, which then led to increased NF-κB-mediated inflammation by degrading IκBα and causing p65 to enter the nucleus. When FAM76B was overexpressed, the translocation of hnRNPA2B1 protein from the nucleus to the cytoplasm was suppressed, and led to the inhibition of NF-κB-mediated inflammation (*Figure 10*). Followed by LPS+IFNγ stimulation, FAM76B and hnRNPA2B1 were found to be partially translocated into the cytoplasm from the nucleus. We speculated that the increased cytoplasmic translocation of FAM76B under inflammatory situation was possibly to balance the increased action of hnRNPA2B1. The detailed mechanism involved needs to be further investigated.

FAM76B is one of the 86 proteins in the human genome that contains stretches of five or more histidines (*Salichs et al., 2009*). Studies have suggested that His-repeats may act as nuclear speckle-targeting signals (*Alvarez et al., 2003*; *Herrmann and Mancini, 2001*; *Salichs et al., 2009*). In our previous study, we confirmed the nuclear speckle localization of both human and mouse FAM76B (*Zheng et al., 2016*). Hence, FAM76B may have functions related to nuclear speckles, such as splicing factor storage and modification (*Salichs et al., 2009*; *McGlincy et al., 2010*). We found in this study that FAM76B interacted with hnRNPA2B1, a heterogeneous nuclear ribonucleoprotein related to mRNA binding and splicing (*Peng et al., 2021*; *Moran-Jones et al., 2005*) that was associated with

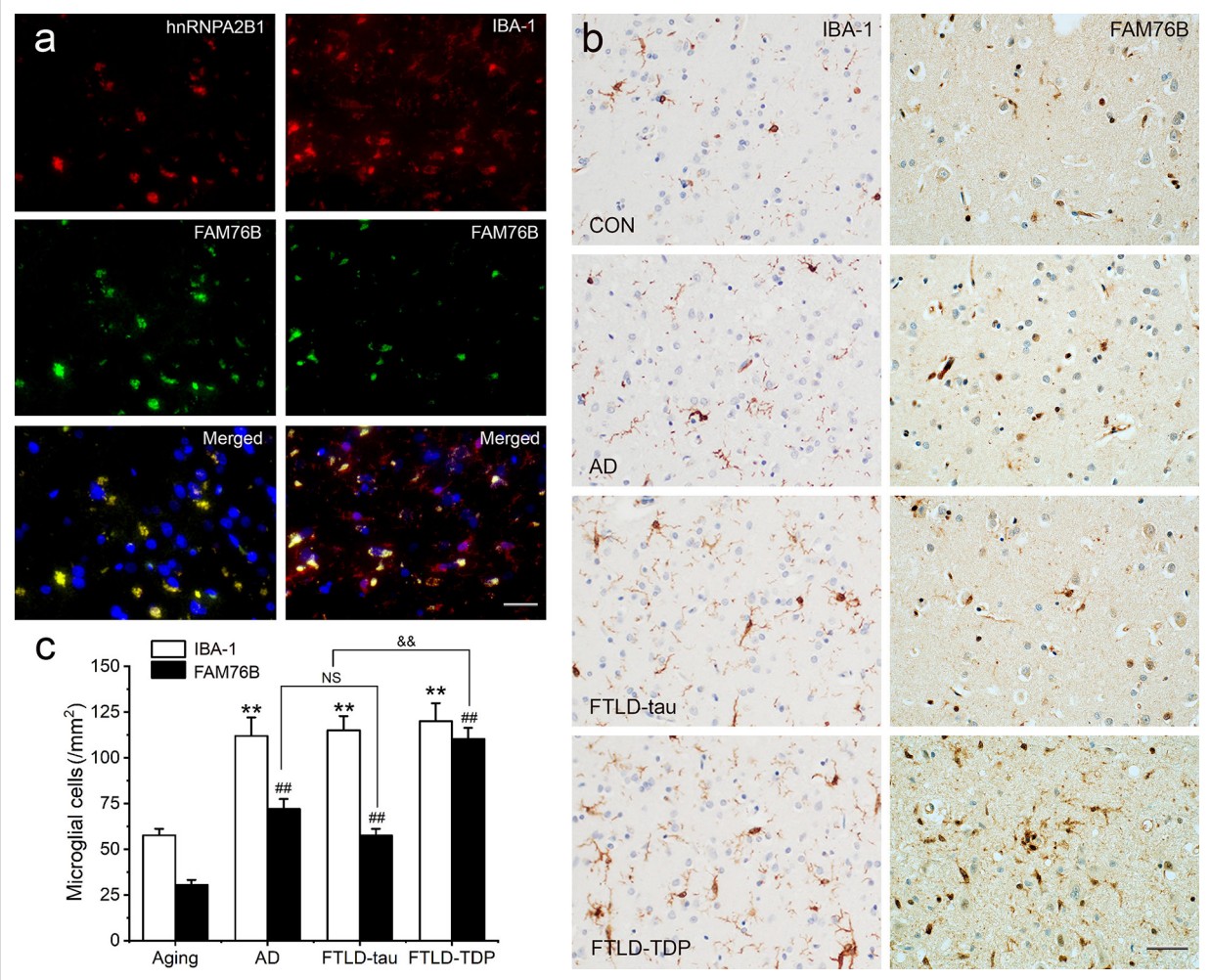

**Figure 9.** Persistent microglial FAM76B expression in neurodegenerative diseases. (**a**) Immunofluorescent staining using the frontal cortex of an frontotemporal lobar degeneration with TAR DNA-binding protein 43 inclusions (FTLD-TDP) patient demonstrated that FAM76B co-localizes with IBA-1, while FAM76B co-localizes with hnRNPA2B1, in the cytoplasm of microglia. Scale bar, 50 μm. (**b**) Immunohistochemical stains revealed that the frontal cortex of Alzheimer's disease (AD), frontotemporal lobar degeneration with tau pathology (FTLD-tau), and FTLD-TDP patients showed increased IBA-1- and FAM76B-positive microglia as compared to the control (CON). This increase in microglial FAM76B expression was more prominent in FTLD-TDP than in AD or FTLD-tau. Scale bar, 50 μm. (**c**) The density of IBA-1- and FAM76B-positive microglia in the frontal cortex of normal aging controls, AD, FTLD-tau and FTLD-TDP patients. Values are mean ± SD; n=5. The densities of IBA-1-positive microglia were compared between different groups by one-way ANOVA followed by the Tukey-honest significant difference (HSD) test (**p<0.01). Similarly, the density of FAM76B-microglia was compared between groups (##p<0.01). The ratios of FAM76B- to IBA-1-microglia densities were compared between different groups by Chi-square goodness-of-fit test (&&p<0.01). n.s., no significance.

inflammation (*Lin and Cao, 2020*; *Chen et al., 2020*; *Coppola et al., 2019*; *Hoffmann et al., 2011*). By in vitro and in vivo experiments, we showed that FAM76B could regulate NF-κB-mediated inflammation via influencing the translocation of hnRNPA2B1.

Neuroinflammation is the response of the CNS to injury and disease and is the common thread that connects brain injuries to neurodegenerative diseases (*Gilhus and Deuschl, 2019*; *Brambilla, 2019*). In TBI, neuroinflammation is one of the most prominent reactions. TBI leads to early resident microglial activation, which is accompanied by local upregulations of TNF-α (*Frugier et al., 2010*; *Csuka et al., 1999*) and IL-6 (*Frugier et al., 2010*; *Perez-Barcena et al., 2011*; *Helmy et al., 2011*). Consistently, we also found increased levels of IL-6 at the contusion site of mouse brains after TBI. In addition, the IL-6 level was further increased by *FAM76B* knockout, indicating the role of the FAM76B-NF-κB pathway in regulating microglial function and driving acute post-traumatic neuroinflammation. TBI can cause persistent neuroinflammation and microglial activation (*Simon et al., 2017*; *Morganti-Kossmann et al., 2019*). Studies of TBI biomarkers in adults with severe TBI have shown that serum

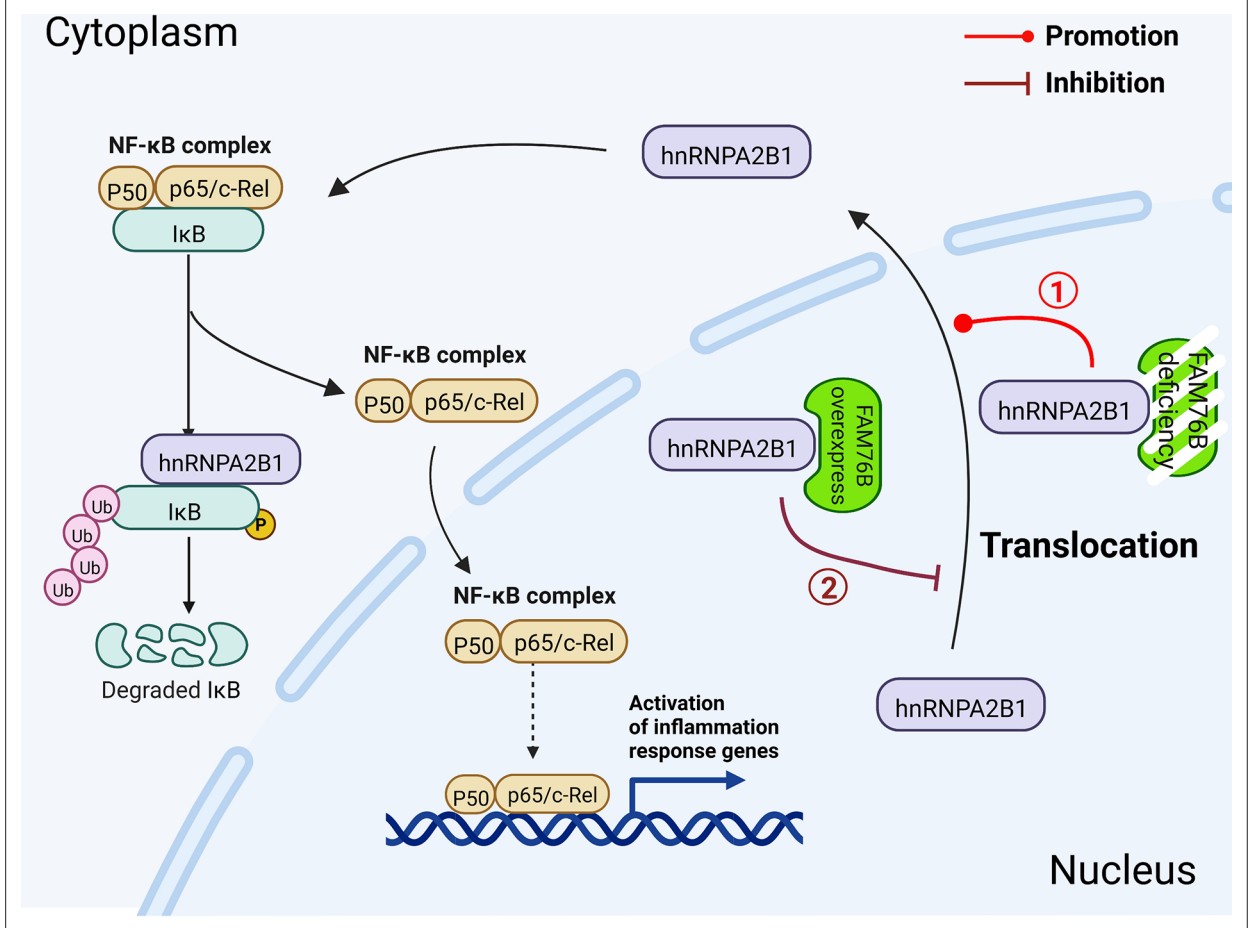

**Figure 10.** Schematic diagram of FAM76B regulating the NF-κB-mediated inflammatory pathway by affecting the hnRNPA2B1 translocation. Under normal conditions, FAM76B could bind to hnRNPA2B1 and make hnRNPA2B1 stay in the nucleus. However, when the expression level of FAM76B was changed, the localization of hnRNPA2B1 was also changed, which then regulated inflammation in immune cells. ① When FAM76B expression was decreased or knocked out in the immune cells, more hnRNPA2B1 proteins translocated into the cytoplasm and led to increased NF-κB-mediated inflammation by the degradation of IKBα and p65's entry into the nucleus. ② When FAM76B expression was increased in the immune cells, FAM76B could trap more hnRNPA2B1 in the nucleus and resulted in less cytoplasmic distribution of hnRNPA2B1 correspondingly, which led to the inhibition of NF-κB mediated inflammation due to the less degradation of IκBα and less entry of P65 into nucleus.

levels of IL-1β, IL-6, and TNF-α are chronically increased (*Simon et al., 2017*). TBI animal models have demonstrated persistently increased numbers of microglia at the margins of the lesion and in the thalamus at 1 year post-injury (*Simon et al., 2017*). Consistent with these findings, we found that in human brain tissue of chronic TBI patients, there was persistent enhanced expression and cytoplasmic distribution of FAM76B in microglia. Persistent microgliosis after TBI correlates with chronic neurodegeneration and dementia development (*Simon et al., 2017*; *Alberici et al., 2018*), which suggests that chronic neuroinflammation may be the mechanism of the neurodegeneration associated with TBI. Furthermore, TBI and neurodegeneration may share a similar neuroinflammatory pathway. In this study, we observed that the enhanced expression and cytoplasmic distribution of FAM76B occurred in both chronic TBI and neurodegenerative disorders, particularly FTLD-TDP, suggesting that the FAM76B-mediated inflammatory pathway might be the common pathway that mediated the neuroinflammation in TBI and FTLD-TDP and that FAM76B-mediated neuroinflammation might be the mechanism by which TBI is linked to neurodegeneration.

In the peripheral tissues, FAM76B deficiency led to increased activation of the NF-κB pathway. In response to an acute proinflammatory stimulus, intraperitoneal LPS administration, *Fam76b*−/− mice showed an interesting phenotype: significantly increased tingible body macrophages in the spleen's white pulp. This finding suggested that FAM76B played a unique role in regulating macrophage function. An overlap between FTLD and autoimmune disease has been noted in the field of

neurodegeneration (*Alberici et al., 2018*). FTLD-associated genetic variants are also linked to auto-immune conditions (*Bright et al., 2019*; *Miller et al., 2013*; *Miller et al., 2016*). Hence, FAM76B dysfunction might be involved in autoimmune processes in FTLD. Though the function of FAM76B in the peripheral tissue is beyond the scope of this paper, it deserves to be further investigated.

In summary, we elucidated, for the first time, a novel function for FAM76B: modulating systemic inflammation and neuroinflammation via influencing the translocation of hnRNPA2B1. We demonstrated the role of FAM76B in the shared neuroinflammatory pathway of TBI and neurodegeneration, particularly in FTLD-TDP. This study may offer important information for the future development of diagnostic biomarkers and immunomodulatory therapeutics for TBI and neurodegeneration, including for FTLD-TDP.

## Materials and methods
### Cell lines
U937 cells (Cat. No: CRL-1593.2) and HEK293 cells (Cat. No: CRL-1573) were both purchased from ATCC. Both cell lines were authenticated using short tandem repeat profiling. They were also tested negative for mycoplasma contamination throughout the experimental period.

### Real-time PCR for gene expression in tissues and cell lines
Total cellular RNA was isolated from cells by TRIzol (Invitrogen, Carlsbad, CA, USA). The cDNA was synthesized using a PrimeScript RT Reagent Kit (Takara Bio, Beijing, China) according to the manufacturer's manual. Gene expression was performed using a real-time PCR kit (Thermo, Rockford, IL, USA) and was normalized to *GAPDH*. The primers used were listed in *Supplementary file 2*.

### Generation of *FAM76B* knockdown and knockout U937 cell lines by lentivirus-mediated Cas9/sgRNA genome editing
An efficient sgRNA (GCAGGGACTGTGGAAACAG) targeting Exon 6 of the human *FAM76B* gene was screened using a T7E1 assay. The primers used for the experiment were listed in *Supplementary file 3*. To generate U937 cell lines with *FAM76B* knockdown, an inducible lentiviral vector was constructed. Briefly, the Bi-Tet-On inducible system, previously established by our lab (*Chen et al., 2015*), and a U6-sgRNA expression cassette were introduced into the lentiviral vector pCDH-CMV-MCS-EF1-*Puro* to generate the inducible lentiviral vector pCDH-Tet-On cassette-U6 sgRNA cassette-EF1-*puro*. Then, Cas9 and the corresponding sgRNAs were cloned into this lentiviral vector to generate the final vector, pCDH-Tet-on-*Cas9*-U6-*FAM76B* sgRNA-EF1-puro. The lentivirus was produced in HEK293T cells. U937 cells were infected by the above lentivirus, screened by puromycin, and then cultured with 2 µg/mL doxycycline for 6 days. The fresh medium containing doxycycline was changed every 3 days to induce expression of Cas9. Then, *FAM76B* knockdown U937 cells were obtained and evaluated by T7E1 assay. Through several rounds of cell cloning by serial dilution of the *FAM76B* knockdown U937 cells, the *FAM76B* knockout U937 cell line was obtained and confirmed by gene sequencing and western blot. In addition, U937 cells were infected by the lentivirus LV-Tet-on-*Cas9*-EF1-*puro* to obtain the control cell line.

### Generation of U937 cell lines overexpressing FAM76B
Human *FAM76B* was inserted into the lentiviral vector pCDH-CMV-MCS-EF1-*Neo* (Addgene, Cambridge, MA, USA) to generate the lentivirus expressing FAM76B. U937 cells were then infected by the produced lentiviruses. After screened by G418, U937 cells overexpressing FAM76B were obtained and confirmed by western blot.

### Mice
All animal studies were performed in accordance with institutional guidelines and with approval by the Institutional Animal Care and Use Committee of Shaanxi Normal University. C57BL/6 mice were obtained from the animal center of Shaanxi Normal University. The mice were maintained in a controlled environment (12/12 hr light/dark cycle, 23±1°C, 55±10% humidity) and given free access to food and water.

## Generation of *FAM76B* gene trap mutant mice and genotyping

Homozygous *FAM76B* knockout (*Fam76b$^{-/-}$*) mice were produced using a commercial service, Texas A&M Institute for Genomic Medicine (College Station, TX, USA), by gene trap mutagenesis techniques. Two germline-competent male chimeras were generated and bred with C57BL/6 female mice. The genotypes of the offspring were determined using tail clip genomic DNA. To detect the WT *Fam76b* allele, PCR was performed using *Fam76b* forward and reverse primers (*Supplementary file 3*). To detect the homozygotes, PCR was performed using *Fam76b* forward and V76 reverse primers (*Supplementary file 3*). The expected size of the fragment from the WT allele was 494 bp. The expected size of the product from intron 1 of the gene trap-containing *Fam76b* gene was 354 bp.

## Isolation of BMDMs

BMDMs were isolated from *Fam76b$^{-/-}$* and WT C57BL/6 mice by flushing the femur with Dulbecco's modified Eagle medium (DMEM) supplemented with 3% fetal bovine serum (FBS). Cells were plated for 4 hr to allow the non-monocytes to adhere to the surface of culture dish. Then, the monocytes in the culture supernatant were collected and centrifuged at 250×*g* and then seeded in DMEM supplemented with 10% FBS, 1% penicillin/streptomycin, 1% L-glutamate, and 10 ng/mL macrophage colony-stimulating factor (Sino Biological Inc, Beijing, China). The fresh medium containing 10 ng/mL macrophage colony-stimulating factor was changed every 2 days to induce differentiation of the cells into macrophages. After 6 days in culture, BMDMs were collected for the following experiments.

## Isolation of mouse embryonic fibroblasts

Embryos from WT and *Fam76b$^{-/-}$* mice were isolated at about E13.5. After the heads, tails, limbs, and most of the internal organs were removed, the embryos were minced and digested in 0.1% (m/v) trypsin for 30 min at 37°C and then centrifuged at 250×*g* to pellet mouse embryonic fibroblasts. The mouse embryonic fibroblasts were cultured in DMEM supplemented with 10% FBS, 1% penicillin/streptomycin, and 1% L-glutamate.

## LPS treatment of mice

For brain injection, LPS (Sigma-Aldrich, St. Louis, MO, USA) (2 μg/μL, 1.5 μL) was injected into the prefrontal cortex using a microsyringe (KD Scientific, model: Legato130) at the following stereotaxic coordinates: 2.60 mm caudal to bregma, 2 mm lateral to the midline, and 1.7 mm ventral to the surface of the dura mater. Vehicle (0.5% methylene blue in PBS or PBS only) was injected in a similar manner into the contralateral prefrontal cortex. For intraperitoneal injection, mice were injected intraperitoneally with LPS (5 μg/g body weight) or with PBS once a day for 2 days before the spleens were harvested for pathologic evaluation.

## Rescue of FAM76B in *FAM76B* knockout U937 cells or BMDMs

For *FAM76B* knockout U937 cells, cells were infected by the lentivirus LV-CMV-*hFAM76B*-EF1-*GFP* or the control lentivirus LV-CMV-MCS-EF1-*GFP* at 5 MOI. Three days after infection, cells were prepared for LPS treatment. For BMDMs from *FAM76B* knockout mice, cells were infected by the lentivirus LV-CMV-*mFAM76B*-EF1-*GFP* or the control lentivirus LV-CMV-MCS-EF1-*GFP* at 10 MOI. Six days after infection, cells were prepared for LPS treatment.

## LPS treatment of cells

To evaluate the effects of FAM76B on the cytokine production of U937 cells or mouse BMDMs, different U937 cell lines or mouse BMDMs were treated with LPS. Briefly, for the impact of FAM76B on cytokine production, *FAM76B* knockdown (*Fam76b KD*) U937 cells were plated in six-well plates (5×10$^5$ cells/well) and treated with 10 ng/mL PMA (Sigma-Aldrich, St. Louis, MO, USA) in association with 10 ng/mL LPS and 20 ng/mL hIFNγ (Sino Biological Inc, Beijing, China) for 48 hr, while *FAM76B* knockout (*Fam76b$^{-/-}$*) U937 cells and FAM76B overexpressed (*Fam76b$^{OE}$*) U937 cells were plated in six-well plates (5×10$^5$ cells/well) and treated with 1 ng/mL PMA for 48 hr, and then were treated with 10 ng/mL LPS and 20 ng/mL hIFNγ for 24 hr. For rescuing FAM76B in the *Fam76b$^{-/-}$* U937 cell line, U937 cells were plated in six-well plates (5×10$^5$ cells/well) and treated with 0.5 ng/mL PMA for 24 hr and then stimulated with 1 ng/mL LPS and 20 ng/mL hIFNγ for 24 hr. For rescuing FAM76B in *Fam76b$^{-/-}$* BMDMs, BMDMs were plated in six-well plates (2×10$^5$ cells/well) and treated with 1 ng/mL LPS and

20 ng/mL mIFNγ (Sino Biological Inc, Beijing, China) for 24 hr. After treatment, cells were harvested in TRIzol for the detection of cytokine expression by real-time PCR.

### IL6 promoter activity assay

Human *IL6* promoter (1868 bp), located in *Homo sapiens* chromosome 7 (NC_000007.14, 22725395-22727262), was obtained by PCR using HEK293 cells genomic DNA as PCR template. The *IL6* promoters were confirmed by gene sequencing and then were inserted into pGL3 basic plasmid to obtain the human *IL6* promoter activity reporter vector pGL3-*IL6* promoter-*Luc*. Human *P50* and *P65* cDNA was obtained by PCR using the MegaMan Human Transcriptome Library as the template. Human *P65* and *P50* were confirmed by gene sequencing and then were inserted into the eukaryotic expression vector pAd5 E1-CMV-MCS to obtain the human P65 and P50 expression vectors pAd5 E1-CMV-*P65* and pAd5 E1-CMV-*P50*, respectively. WT HEK293 cells and *FAM76B* knockout (*Fam76b$^{-/-}$*) HEK293 cells were plated into 24-well plates at a density of $2 \times 10^5$ cells per well. The next day, the promoter activity reporter vector pGL3-*IL6* promoter-*Luc* (200 ng) and the Renilla luciferase expression vector pRL-CMV (50 ng) as the internal reference plasmid were co-transfected with the P65 and P50 expression vector pAd5 E1-CMV-*P65/P50* (each 300 ng) or the control vector pAd5 E1-CMV-MCS (600 ng) into the cells using X-tremeGENE HP Reagent (Roche, Indianapolis, IN, USA) according to the manufacturer's protocol. 48 hr later, the cells were collected for a luciferase activity assay using a dual-luciferase assay kit (Promega, Madison, WI, USA). The normalized luciferase activity was obtained by using the formula: Normalized luciferase value = Fly luciferase value/Renilla luciferase value.

Meanwhile, the human *IL6* promoter was inserted into the upstream of luciferase in the lentivirus vector pCDH-*luc*-EF1-*Neo* to obtain the vector pCDH-*IL6* promoter-*luci*-EF1-*Neo*. Then, the WT or *Fam76b$^{-/-}$* U937 cells were infected by the above lentivirus. After selected by G418, WT and *Fam76b$^{-/-}$* U937 cells were seeded into 24 wells ($2 \times 10^5$ cells per well) and incubated with different concentrations of LPS for 36 hr. The cells were then collected for a luciferase activity assay.

Two kinds of human NF-κB binding motif (motif 1: GGGAATTTCC, motif 2: GGGATTTTCC) were also synthesized and inserted into the upstream of miniCMV promoter in the lentivirus vector pCDH-miniCMV-*luci*-EF1-*Neo* to obtain the vector pCDH-NF-κB binding motif 1(or 2)-miniCMV-*luci*-EF1-*Neo*. Then, the WT or *Fam76b$^{-/-}$* U937 cells were infected by the above two kinds of lentivirus. After selected by G418, WT and *Fam76b$^{-/-}$* U937 cells were seeded into 24 wells ($2 \times 10^5$ cells per well) and incubated with different concentrations of LPS for 36 hr. Then the cells were collected for a luciferase activity assay.

### Flow cytometry

Spleens from 5-month-old WT and *Fam76b$^{-/-}$* mice were pressed through an 80 μm mesh with a syringe plunger to acquire single-cell suspensions. Red blood cells were removed from single-cell suspensions using red blood cell lysis buffer (Shanghai Yeasen Biotechnology, Shanghai, China). The cells were then washed twice and resuspended in 200 μL wash buffer for final flow cytometric analysis. Fluorescent-labeled antibodies used for flow cytometry were listed in *Supplementary file 4*, and flow cytometry staining was performed according to the manufacturer's instructions using propidium iodide solution (BioLegend, San Diego, CA, USA) to exclude dead cells.

### Western blot

Cells were lysed by RIPA buffer. Cell lysis was subjected to SDS-PAGE and subsequently blotted onto methanol pretreated polyvinylidene difluoride (PVDF) membranes. The PVDF membranes were incubated with the primary antibody overnight at 4°C. Membranes were washed and incubated with corresponding horseradish peroxidase-conjugated secondary antibody. The membranes were visualized using enhanced chemiluminescence western blot detection reagents (Thermo Fisher, Waltham, MA, USA) in a chemical luminescence imaging apparatus according to the manufacturer's protocol. Primary antibodies used for western blot were listed in *Supplementary file 4*.

### Experimental TBI by CCI

Experimental TBI was performed using the CCI model, as described (*Zheng et al., 2022*). Briefly, mice were anesthetized with 1.5% pentobarbital sodium at a dose of 50 mg/kg. The mice were fixed in a stereotactic frame and subjected to a craniotomy (5 mm diameter) in the right parietal region using a

motorized drill. The brain was exposed to a pneumatic impactor (brain injury device TBI-68099, RWD, China), and TBI was produced using the following parameters: diameter impactor tip, 3 mm diameter metal tip; velocity, 3.3 m/s; duration, 0.1 s; depth of penetration, 2 mm. After the wound was sutured, an electric heater was used to maintain the animals' body temperature until they were completely awake and able to move freely, which occurred approximately 1–2 hr after the injury. Buprenorphine was diluted in 0.9% NaCl to a concentration of 0.01 mg/mL, and a 0.1 mg/kg dose was administered subcutaneously, which provided 72 hr of sustained post-operative analgesia. Sham-operated WT and *Fam76b* knockout mice, used as controls, were treated the same as the CCI-treated mice, except for the craniotomy and CCI.

## Cytokine expression of mouse brains by ELISA and real-time PCR

The mice were perfused transcardially with ice-cold PBS, and prefrontal cortex tissues were carefully removed on ice. The tissues were weighed quickly, and the total protein extracts of the prefrontal cortex were obtained by homogenization in mammalian cell lysis reagent (Pioneer Biotechnology, Shanghai, China) with a protease inhibitor mixture (Roche, Indianapolis, IN, USA). The levels of IL-6 and TNF-α were quantified using a QuantiCyto ELISA kit (NeoBioscience, Shenzhen, Guangdong, China). Real-time PCR was performed on tissues from the prefrontal cortices of mice (see Materials and methods above).

## IP-MS and data analysis

U937 cells were infected by FAM76B-Strep tagII or control Strep tagII-expressing lentiviruses, followed by stable cell line screening. The total protein was extracted, purified with Strep-Tactin beads (QIAGEN, Düsseldorf, Germany), and sent to Shanghai Bioprofile Technology for mass spectrometry sequencing. After high-performance liquid chromatography and mass spectrometry analysis, the MS data were analyzed using MaxQuant software version 1.6.0.16. MS data were searched against the UniProtKB Rattus norvegicus database (36,080 total entries, downloaded on August 14, 2018). Trypsin was selected as the digestion enzyme. A maximum of two missed cleavage sites and mass tolerances of 4.5 ppm for precursor ions and 20 ppm for fragment ions were defined for the database search. Carbamidomethylation of cysteines was defined as a fixed modification, while acetylation of the protein N-terminal and oxidation of methionine were set as variable modifications for database searching. The database search results were filtered and exported with a <1% false discovery rate at the peptide-spectrum-matched level and the protein level.

## Co-immunoprecipitation

Human cDNAs of full-length *hnRNPA2B1* and different domains of hnRNPA2B1, including *RRM1, RRM2, and RGD*, were constructed by PCR using pGEMT/*hnRNPA2B1* (kept in the Xia lab) as the template and the primers listed in *Supplementary file 3*. HEK293 cells were co-transfected with FAM76B-StreptagII and hnRNPA2B1-Flag-expressing vectors. At 48 hr after transfection, approximately 500 μg of protein extracts prepared from these cells were incubated with Strep-Tactin beads at 4°C for 3 hr. The bound protein was examined by western blot with anti-Flag or anti-FAM76B antibodies. Co-immunoprecipitation of hnRNPA2B1 and IκB-flag was performed similarly, except that protein A/G-Sepharose beads (Thermo, Rockford, IL, USA) charged with anti-hnRNPA2B1 antibody or mouse normal serum were used.

## Confocal microscopy

HEK293 cells expressing the fusion proteins eGFP-FAM76B and mCherry-tagged hnRNPA2B1 were visualized using a Leica TCS-SP8 confocal microscope (Leica Microsystems Inc, Shanghai, China).

## Human tissues

Autopsy brain tissues from patients who had had either a TBI or dementia were collected under an IRB-approved protocol at the University of Utah. Paraformaldehyde-fixed, paraffin-embedded human brain samples from 23 dementia cases (*Supplementary file 5*) were acquired from the Neuropathology Core of Northwestern University's Center for Cognitive Neurology and Alzheimer's disease (Chicago, IL, USA). Demographic and neuropathologic data for these cases were presented in *Supplementary file 5*. Pathologic characterization was made by board-certified neuropathologists blinded

to case identity and following consensus criteria (*Mackenzie et al., 2009*; *Cairns et al., 2007*; *Mackenzie et al., 2010*; *McKhann et al., 2001*).

## Immunofluorescence and immunohistochemistry

For immunofluorescence, unstained slides were deparaffinized, followed by antigen retrieval using Diva Decloaker buffer (Biocare Medical, Pacheco, CA, USA). The slides were then blocked with 1% bovine serum albumin and 0.1% Triton in PBS for 1 hr, before applying primary antibody overnight. After overnight incubation, secondary antibody was applied for 45 min, before being washed and coverslipped. The following primary antibodies were used: anti-FAM76B (monoclonal, 1:1000, in-house; *Zheng et al., 2016*), IBA-1 (goat polyclonal, 1:1000, Abcam, Boston, MA, USA), or hnRNPA2B1 (rabbit polyclonal, 1:1000, Abcam, Boston, MA, USA) antibodies. Secondary antibodies included donkey anti-rabbit, donkey anti-goat, and donkey anti-mouse antibodies (1:500; Abcam, Boston, MA, USA). Cultured cells were stained in a similar manner except without being deparaffinized and without the antigen retrieval step. For immunohistochemistry, unstained slides were deparaffinized, followed by antigen retrieval using Diva Decloaker buffer. The slides were then quenched with 10% $H_2O_2$ in methanol and blocked as described above. The primary antibodies, including monoclonal anti-FAM76B (monoclonal, 1:1000, in-house; *Zheng et al., 2016*) or IBA-1 (goat polyclonal, 1:1000, Abcam, Boston, MA, USA) antibodies, were then applied overnight, followed by biotinylated secondary antibody (1:500, Abcam, Boston, MA, USA). Signal detection was performed using a VECTASTAIN Elite ABC kit and DAB (Vector Laboratories, Burlingame, CA, USA). The stains were reviewed using an Olympus BX53 microscope (Tokyo, Japan). Representative images were taken with an Olympus DP74 camera, and cellSens Dimension software was used for brightness and contrast adjustment and image cropping.

## Microglial quantification

Microglia were counted using a 40× objective with a grid ($250\times250$ µm$^2$) in a minimum of five microscopic grid fields in the area of interest per slide. Results were given as mean objects per unit area (mm$^2$). Microglial cells that had stained cytoplasmic processes and contained a nucleus in the plane of the section were counted.

## Statistics

Group effects were evaluated using unpaired t-tests (Mann-Whitney), paired t-tests, and one-way ANOVA followed by the Tukey honest significant difference test. The difference in the ratio of the microglia density was evaluated by Chi-square goodness-of-fit tests. All statistical analyses were performed using GraphPad Prism software (version 4.01). Differences between the means were considered significant at $p<0.05$.

## Study approval

All animal studies were performed in accordance with institutional guidelines and with approval by the Institutional Animal Care and Use Committee of Shaanxi Normal University (SNNU 2019-0128). Ethical permit of the use of the samples of human brain autopsy specimens was granted by the ethics committee of Shaanxi Normal University (SNNU 2019-0026).

# Acknowledgements

This work was supported by research grants to HX from the National Natural Science Foundation of China (No. 81773265), Key Research and Development Plan of Shaanxi Province (No. 2018SF-106), and a grant to XZ from the Natural Science Foundation of Shaanxi Province, China (No. 2023-JC-YB-642), grants to HX from the Fundamental Research Funds for the Central Universities (No. GK202007023 and GK202107018).

## Additional information

### Funding

| Funder | Grant reference number | Author |
| --- | --- | --- |
| National Natural Science Foundation of China | 81773265 | Haibin Xia |
| Key Research and Development Plan of Shaanxi Province | 2018SF-106 | Haibin Xia |
| Natural Science Foundation of Shaanxi Province | 2023-JC-YB-642 | Xiaojing Zheng |
| The Fundamental Research Funds for the Central Universities | GK202007023 and GK202107018 | Haibin Xia |

The funders had no role in study design, data collection and interpretation, or the decision to submit the work for publication.

### Author contributions

Dongyang Wang, Writing - original draft, Writing - review and editing; Xiaojing Zheng, Methodology, Writing - original draft; Lihong Chai, Jiuling Zhu, Yanqing Li, Investigation, Methodology; Junli Zhao, Peiyan Yang, Methodology; Qinwen Mao, Conceptualization, Methodology, Writing - review and editing; Haibin Xia, Conceptualization, Supervision, Writing - review and editing

### Author ORCIDs

Dongyang Wang http://orcid.org/0000-0001-9710-636X
Xiaojing Zheng http://orcid.org/0009-0008-5510-5041
Haibin Xia http://orcid.org/0000-0002-2038-5759

### Ethics

All animal studies were performed in accordance with institutional guidelines and with approval by the Institutional Animal Care and Use Committee of Shaanxi Normal University (SNNU 2019-0128). Ethical permit of the use of the samples of human brain autopsy specimens was granted by the ethics committee of Shaanxi Normal University (SNNU 2019-0026).

### Decision letter and Author response

Decision letter https://doi.org/10.7554/eLife.85659.sa1
Author response https://doi.org/10.7554/eLife.85659.sa2

## Additional files

### Supplementary files

- Supplementary file 1. Proteins that interact with FAM76B and their interacting scores and ranks.
- Supplementary file 2. Primers used for real-time PCR.
- Supplementary file 3. Other primers used in the study.
- Supplementary file 4. Antibodies used in the study.
- Supplementary file 5. Patient demographics.
- MDAR checklist

### Data availability

The mass spectrometry proteomics data have been deposited to the ProteomeXchange Consortium (http://proteomecentral.proteomexchange.org) via the iProX partner repository [1] with the dataset identifier PXD037539. ([1] Ma J, et al. (2019) iProX: an integrated proteome resource. Nucleic Acids Res, 47, D1211-D1217).

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
