## [Editor Report]

This fundamental study identifies the protein FAM76B as a regulator of inflammation and provides evidence for its mechanism of action. FAM76B previously had no clear cellular function but the solid experimental methods described in this study established it as a negative regulator of inflammation. Additional investigation into the FAM76B protein is needed to better understand its role in physiological and pathological conditions associated with inflammation.

---

## [Decision Letter]

**Decision letter after peer review:**

Thank you for submitting your article "FAM76B regulates NF-κB-mediated inflammatory pathway by influencing the translocation of hnRNPA2B1" for consideration by *eLife*. Your article has been reviewed by 2 peer reviewers, and the evaluation has been overseen by a Reviewing Editor and Mone Zaidi as the Senior Editor. The reviewers have opted to remain anonymous.

Essential revisions:

1) Section 3.2 "FAM76B regulates the NF-κB pathway by influencing the translocation of hnRNPA2B1" on pages 13-14 requires additional data to support the title or revise to discuss whether the possible role of increased cytoplasmic translocation of FAM76B under inflammatory situation is to balance the increased action of hnRNPA2B1, and not to promote the activation of NF-κB pathway. Overexpression of FAM76B with and without stimulation of LPS and IFNγ could be informative. Additionally, consider altering NFKB or hnRNPA2B1 in both WT and FAM76B KO cells in response to inflammatory stimuli such as LPS or TNFa to strengthen conclusion.

2) The columns in Figure 9 different magnifications which exaggerates effect, please provide same magnification for both groups. Also, is the n=5 for figure 9 from 5 different brains? Or 5 different regions of one section? From methods, it seems it is 5 regions of one brain section. This should be removed or revised to clearly note that it is comparing two patients.

3) The mouse data do not support a role of FAM76B in TBI, it only shows that an inflammatory trigger increased Iba1 and cytokine expression. Claims for a role of FAM76B for TBI should be revised.

4) The custom antibody reacts with mouse FAM76B based on 2016 paper. Which cells express FAM76B in the mouse brain? Is there evidence that microglia express it in mice? Or is increased Iba1 an indirect effect? The authors should consider a figure showing the colocalization of FAM76B with microglia in the mouse brain to support the increased Iba1 is direct effect of FAM76B effects in microglia.

5) The heading for 3.4 should be revised. There is no evidence to support that FAM76B plays a role in neurodegeneration or TBI. Authors should remove mention of FAM76B playing a role in neurodegeneration. If anything, it seems it is increased in Iba1 cells but wouldn't that possibly decrease the inflammatory state of microglia? For example, point d in the first paragraph of Discussion: what is the evidence that TBI shows "activation, evolution and persistence of FAM76B-NFkB mediated neuroinflammation during repair process" in human brain? Similar concern for point e. Authors should focus on the data provided that supports causal not correlative role. The finding that FAM76B can regulate neuroinflammatory responses is the primary takeaway from the current study with the data presented. Overall, the human data is interesting but not adequate to support a claim of FAM76B having a contribution to neurodegeneration or TBI.

*Reviewer #1 (Recommendations for the authors):*

The manuscript from Wang et al., first demonstrated that FAM76B inhibited the NF-κB-mediated inflammation by modulating the cytosol translocation of hnRNPA2B1, which is a potentially important finding about inflammation in neurodegenerative diseases which not much is known. There is a need in the field to better understand the underlying mechanism. Overall, the manuscript is well-written and easy to follow, and the study is focused on an important question in the roles and mechanism of FAM76B in regulation of inflammation. This work is an important step forward.

1. FAM76B was described as a negative regulator of inflammation in this work, could the author discuss whether it had a directly function on NFkB or its upstream signaling?

2. In figure 4, if the authors could provide evidence of the alteration of NKkB, hnRNPA2B1 in both WT and FAM76B KO cells in response to inflammatory stimuli such as LPS or TNFa, will strength the conclusion.

*Reviewer #2 (Recommendations for the authors):*

A novel role of FAM76B in regulation of NF-κB-mediated inflammation, specially in neuroinflammation by using gene knockdown and knockout cell line and animal models was clearly and logically laid out. However, the major concern is that it is still unclear how the FAM76B regulates/or affects the cytoplasmic translocation of hnRNPA2B1 in brain cells after a variety of injuried stimuli and thereby influencing the NF-κB activation.

1. Results of 3.2 "FAM76B regulates the NF-κB pathway by influencing the translocation of hnRNPA2B1" on pages, 13-14, are unclear. First, data from U937 cells with FAM76B gene knockout upon the LPS stimulation showed that inhibition of FAM76B increased the activity of the IL-6 promotor, NF-κB binding motifs (Figure 2), cytosolic hnRNPA2B1 levels, phosphates IKBa and IKBß and nuclear activation of p65 (Figure 4). However, Figure 4h showed that both FAM76B and hnRNPA2B1 were found to be partially translocated into the cytoplasm from the nucleus in the M1-like macriphage cells after LPS and IFNγ stimulation. Thus, the authors concluded that"FAM76B could promote NF-κB activation by affecting the translocation of hnRNPA2B1", which was not supported but conflicted by Figures2, and 4a -g. This statement needs to be further examined in U937 cells with over expression of FAM76B with or without LPS and IFNγ stimulation. At least, the authors may need to discuss whether the possible role of increased cytoplasmic translocation of FAM76B under inflammatory situation is to balance the increased action of hnRNPA2B1, but not to promote the activation of NF-κB pathway.

---

## [Author Response]

Essential revisions:1) Section 3.2 "FAM76B regulates the NF-κB pathway by influencing the translocation of hnRNPA2B1" on pages 13-14 requires additional data to support the title or revise to discuss whether the possible role of increased cytoplasmic translocation of FAM76B under inflammatory situation is to balance the increased action of hnRNPA2B1, and not to promote the activation of NF-κB pathway. Overexpression of FAM76B with and without stimulation of LPS and IFNγ could be informative. Additionally, consider altering NFKB or hnRNPA2B1 in both WT and FAM76B KO cells in response to inflammatory stimuli such as LPS or TNFa to strengthen conclusion.

(1) Thanks for your suggestion. We are very sorry for the incorrect conclusion "FAM76B could promote NF-κB activation by affecting the translocation of hnRNPA2B1" in Section 3.2 "FAM76B regulates the NF-κB pathway by influencing the translocation of hnRNPA2B1" on pages 16 in the original manuscript. It was made by our mistake. We have corrected this. The right statement is “FAM76B could inhibit NF-κB activation by affecting the translocation of hnRNPA2B1.” which is (See the revised manuscript on page 7, lines 151-152). (2) According to your suggestion, we revised discussion part about increased cytoplasmic translocation of FAM76B under inflammatory situation (See Page 12, lines 284-288 in the revised manuscript). (3) According to your suggestion, we included the results of U937 cells with the overexpression of FAM76B treated with PMA only or with PMA followed by LPS+IFNγ stimulation in revised manuscript on Page 6, lines 137-146. The results indicated that overexpression of FAM76B could downregulate several inflammatory cytokines, such as IL-6, PTGS2 and TNF-α, indicating that FAM76B had an anti-inflammatory effect (See revised Figure 4—figure supplement 1), and inhibit the translocation of hnRNPA2B1 from nucleus to cytoplasm in U937 cells followed by LPS and IFNγ stimulation compared to the control group (See revised Figure 4—figure supplement 2 and 3). (4) According to your suggestion, we detected the alteration of hnRNPA2B1 in both WT and FAM76B KO cells in response to PMA/ LPS+IFNγ stimulation and found the increased cytoplasmic translocation of hnRNPA2B1 in *Fam76b^-/-^* U937 cells. The results were added in the manuscript on Page 6, lines 135-136 (See revised Figure 4d-f). Considering the logicality of the results from Figure 4, the original Figure 4h was moved and renamed revised Figure 4—figure supplement 4. Besides, in the study, we mainly focused the effect of FAM76B on the translocation of the hnRNPA2B1. But we also had detected the alteration of the protein P65, an important subunit of NF-κB, in both WT and FAM76B KO cells in response to LPS+IFNγ stimulation and found an increased nuclear translocation of p65 in *Fam76b^-/-^* U937 cells stimulated with PMA followed by incubation with LPS and hIFNγ, indicating the activation of NF-κB mediated inflammation (See the revised Figure 4i and j, and the revised manuscript on Page 7, lines 149-151).

2) The columns in Figure 9 different magnifications which exaggerates effect, please provide same magnification for both groups. Also, is the n=5 for figure 9 from 5 different brains? Or 5 different regions of one section? From methods, it seems it is 5 regions of one brain section. This should be removed or revised to clearly note that it is comparing two patients.

We have provided new figures with same magnification for both groups in Figure 9 (See revised Figure 9). Besides, in Figure 9, pathological samples were from 5 different patient brains, the basic information of the patients was shown in Supplementary File 5. In fact, this immunohistochemical experiment was performed in 4 different group of patients (normal aging controls, and AD, FTLD-tau, and FTLD-TDP patients), including 5 patients in each group, and microglia for every patient were counted in a minimum of five microscopic grid fields in the area of interest per slide as mentioned in methods.

3) The mouse data do not support a role of FAM76B in TBI, it only shows that an inflammatory trigger increased Iba1 and cytokine expression. Claims for a role of FAM76B for TBI should be revised.

We have claimed a role of FAM76B for TBI on Page 9, Lines 213-220 in the revised manuscript.

4) The custom antibody reacts with mouse FAM76B based on 2016 paper. Which cells express FAM76B in the mouse brain? Is there evidence that microglia express it in mice? Or is increased Iba1 an indirect effect? The authors should consider a figure showing the colocalization of FAM76B with microglia in the mouse brain to support the increased Iba1 is direct effect of FAM76B effects in microglia.

We have detected the expression of FAM76B in mouse brain and found the colocalization of FAM76B with microglia using immunofluorescence staining. And the result has been added on Page 8, Lines 187-188 (See revised Figure 7—figure supplement 1).

5) The heading for 3.4 should be revised. There is no evidence to support that FAM76B plays a role in neurodegeneration or TBI. Authors should remove mention of FAM76B playing a role in neurodegeneration. If anything, it seems it is increased in Iba1 cells but wouldn't that possibly decrease the inflammatory state of microglia? For example, point d in the first paragraph of Discussion: what is the evidence that TBI shows "activation, evolution and persistence of FAM76B-NFkB mediated neuroinflammation during repair process" in human brain? Similar concern for point e. Authors should focus on the data provided that supports causal not correlative role. The finding that FAM76B can regulate neuroinflammatory responses is the primary takeaway from the current study with the data presented. Overall, the human data is interesting but not adequate to support a claim of FAM76B having a contribution to neurodegeneration or TBI.

Thanks for your suggestions. According to your suggestion, we revised the heading for this Section (See the revised manuscript, on Page 9, lines 221-222). Besides, point d and e in the first paragraph of Discussion have also been revised (See the revised manuscript on Page 12, Lines 272-276).

Reviewer #1 (Recommendations for the authors):The manuscript from Wang et al., first demonstrated that FAM76B inhibited the NF-κB-mediated inflammation by modulating the cytosol translocation of hnRNPA2B1, which is a potentially important finding about inflammation in neurodegenerative diseases which not much is known. There is a need in the field to better understand the underlying mechanism. Overall, the manuscript is well-written and easy to follow, and the study is focused on an important question in the roles and mechanism of FAM76B in regulation of inflammation. This work is an important step forward. However, some of the details were missed in the MS, which I've tried to outline below.

1. FAM76B was described as a negative regulator of inflammation in this work, could the author discuss whether it had a directly function on NFkB or its upstream signaling?

According to your suggestion, we added the discussion about the mechanism of FAM76B inhibiting the inflammation in the discussion part in the revised Figure 10 and revised manuscript (See Page 12, lines 282-288). The Figure legend of Figure 10 was also revised.

2. In figure 4, if the authors could provide evidence of the alteration of NFkB, hnRNPA2B1 in both WT and FAM76B KO cells in response to inflammatory stimuli such as LPS or TNFa, will strength the conclusion.

Thank you for your good suggestion. According to your suggestion, we detected the alteration of hnRNPA2B1 in both WT and FAM76B KO cells in response to LPS+IFNγ stimulation and found the increased cytoplasmic translocation of hnRNPA2B1 in *Fam76b^-/-^* U937 cells. The results were added in the manuscript on Page 6, lines 135-136 (See revised Figure 4d-f). Besides, we had detected the alteration of the protein P65, an important subunit of NF-κB, in both WT and FAM76B KO cells in response to LPS+IFNγ stimulation and found an increased nuclear translocation of p65 in *Fam76b^-/-^* U937 cells stimulated with PMA followed by incubation with LPS and hIFNγ, indicating the activation of NF-κB mediated inflammation (See revised Figure 4i and j and the revised manuscript on Page 7, lines 149-151). We believe that all these results support our conclusion of the paper.

Reviewer #2 (Recommendations for the authors):A novel role of FAM76B in regulation of NF-κB-mediated inflammation, specially in neuroinflammation by using gene knockdown and knockout cell line and animal models was clearly and logically laid out. However, the major concern is that it is still unclear how the FAM76B regulates/or affects the cytoplasmic translocation of hnRNPA2B1 in brain cells after a variety of injuried stimuli and thereby influencing the NF-κB activation.1. Results of 3.2 "FAM76B regulates the NF-κB pathway by influencing the translocation of hnRNPA2B1" on pages, 13-14, are unclear. First, data from U937 cells with FAM76B gene knockout upon the LPS stimulation showed that inhibition of FAM76B increased the activity of the IL-6 promotor, NF-κB binding motifs (Figure 2), cytosolic hnRNPA2B1 levels, phosphates IKBa and IKBß and nuclear activation of p65 (Figure 4). However, Figure 4h showed that both FAM76B and hnRNPA2B1 were found to be partially translocated into the cytoplasm from the nucleus in the M1-like macriphage cells after LPS and IFNγ stimulation. Thus, the authors concluded that"FAM76B could promote NF-κB activation by affecting the translocation of hnRNPA2B1", which was not supported but conflicted by Figures2, and 4a -g. This statement needs to be further examined in U937 cells with over expression of FAM76B with or without LPS and IFNγ stimulation. At least, the authors may need to discuss whether the possible role of increased cytoplasmic translocation of FAM76B under inflammatory situation is to balance the increased action of hnRNPA2B1, but not to promote the activation of NF-κB pathway.

Thank you for your good suggestion.

(1) Thank you for your suggestion. We are very sorry for the incorrect conclusion "FAM76B could promote NF-κB activation by affecting the translocation of hnRNPA2B1" in Section 3.2 "FAM76B regulates the NF-κB pathway by influencing the translocation of hnRNPA2B1" on pages 16 in the original manuscript. It was made by our mistake. We have corrected this. The right statement is “FAM76B could inhibit NF-κB activation by affecting the translocation of hnRNPA2B1.” which is (See the revised manuscript on page 7, lines 151-152). (2) According to your suggestion, we revised discussion part about increased cytoplasmic translocation of FAM76B under inflammatory situation (See Page 12, lines 284-288 in the revised manuscript). (3) According to your suggestion, we included the results of U937 cells with the overexpression of FAM76B treated with PMA only or with PMA followed by LPS+IFNγ stimulation in revised manuscript on Page 6, lines 137-146. The results indicated that overexpression of FAM76B could downregulate several inflammatory cytokines, such as IL-6, PTGS2 and TNF-α, indicating that FAM76B had an anti-inflammatory effect (See revised Figure 4—figure supplement 1), and inhibit the translocation of hnRNPA2B1 followed by LPS and IFNγ stimulation compared to the control group (See revised Figure 4—figure supplement 2 and 3). (4) We added the discussion about the mechanism of FAM76B inhibiting the inflammation in the discussion part in the revised Figure 10 and revised manuscript (See Page 12, lines 282-288).